**The simulation of mineral dust in the United Kingdom Earth System Model UKESM1.**

**Stephanie Woodward[1], Alistair A. Sellar[1], Yongming Tang[1], Marc Stringer[2], Andrew Yool[3], Eddy Robertson[1] and Andy Wiltshire[1].**

[1]Met Office Hadley Centre, Exeter, UK
[2]Department of Meteorology, University of Reading, Reading, UK
[3]National Oceanography Centre, Southampton, UK

*Correspondence to*: Stephanie Woodward (stephanie.woodward@metoffice.gov.uk)

**Abstract.** Mineral dust plays an important role in earth system models, being linked to many components: atmospheric wind-speed, precipitation and radiation, surface vegetation cover and soil properties, and oceanic biogeochemical systems. In this paper the dust scheme in the first configuration of the United Kingdom Earth System Model UKESM1 is described,

and simulations of dust and its radiative effects are presented and compared with results from the parallel coupled atmosphere-ocean general circulation model (GCM) HadGEM3-GC3.1. Not only changes in the driving model fields but also changes in the dust size distribution are shown to lead to considerable differences to the present-day dust simulations and to projected future changes. UKESM1 simulations produce a present-day top of the atmosphere (ToA) dust direct radiative effect (DRE - defined as the change in downward net flux directly due to the presence of dust) of 0.086 Wm$^{-2}$ from

a dust load of 19.5 Tg. Under climate change pathways these values decrease considerably: in the 2081–2100 mean of Shared Socioeconomic Pathway SSP5-8.45 ToA DRE reaches 0.048 W m$^{-2}$ from a load of 15.1 Tg. In contrast, in HadGEM3-GC3.1 the present-day values of -0.296 W m$^{-2}$ and 15.0 Tg are almost unchanged, at -0.289 W m$^{-2}$ and 14.5 Tg in the 2081–2100 mean. The primary mechanism causing the differences in future dust projections is shown to be the vegetation response, which dominates over the direct effects of warming in our models. Though there are considerable

uncertainties associated with any such estimates, the results presented demonstrate both the importance of the size distribution for dust modeling, and also the necessity of including earth system processes such as interactive vegetation in dust simulations for climate change studies.

## 1 Introduction

Dust is an important component of the earth system being linked to the atmosphere, aerosols, surface, and terrestrial and oceanic biospheres (Carslaw et al., 2010). Many factors control the dust lifecycle, including vegetation cover, soil moisture, windspeed and precipitation; hence dust is potentially sensitive to changes in climate through multiple mechanisms. Dust, in turn, affects climate via radiative processes in the atmosphere, via interaction with clouds and other aerosol species, by changing surface albedo and as a source of nutrients for oceanic and terrestrial biogeochemical cycles.

Climate models have long included dust schemes (e.g. Tegen and Miller, 1998; Woodward, 2001; Miller et al., 2006; Kok et al., 2014), which typically allowed dust emission, transport and deposition to be modelled, together with the direct radiative effects of atmospheric dust. Some attempts were made to include the impact of the vegetation response on dust (Tegen et al., 2002; Mahowald et al., 2006), though as the vegetation was not simulated interactively, feedbacks between vegetation and climate were excluded. The advent of earth system models (Collins et al., 2011; Watanabe et al., 2011), which include

interactive vegetation and ocean biogeochemical schemes together with other components, has extended the possibilities for the investigation of interactions between dust and other systems. This also allows for the impact of the responses of earth system components on the dust to be included in future climate projections.

Here we employ the state-of-the-art UKESM1 model (Sellar et al.,2019) to explore the behaviour of dust within the earth system in the present day and under possible future development pathways. UKESM1 is built around the HadGEM3-GC3.1

coupled ocean-atmosphere model (Kulbrodt et al., 2018; Williams et al., 2018), and we have made use of this to run parallel experiments with the coupled and earth system models, in order to investigate the importance of earth system processes to the dust simulation. Both models use the same dust emission scheme, though the tuning parameters were adjusted for UKESM1. We include particle sizes up to 63 μm diameter. Recent studies have suggested that the correct representation of super-coarse particles is important for the simulation of dust radiative effects (Ryder et al., 2019; Kok et al.,2017).

In this paper we give a brief introduction to UKESM1 and describe the dust scheme and the experiments used before assessing the present-day simulation of dust load, size distribution, aerosol optical depth (AOD) and deposition against observations, and we also present the simulated dust DRE results. We then compare the UKESM1 simulations with parallel results from HadGEM3-GC3.1 and investigate the causes of the differences: in particular the consequences of including interactive vegetation and other earth system processes and the effects of the changes to the dust scheme settings, including

the impacts of re-tuning on load and size distribution and of excluding emissions from seasonal sources. In the next section we investigate the response of dust to potential climate change as driven by two Shared Socio-Economic Pathways and consider the mechanisms involved. Finally we present a discussion of the results and summarise our main findings.

## 2 UKESM1

UKESM1 is the latest generation UK earth system model built around the HadGEM3-GC3.1 coupled Atmosphere-Ocean

GCM (Kulhbrodt et al., 2018; Williams et al., 2017) combined with the MEDUSA biogeochemical model (Yool et al.,

2013).  The interface between these is provided by the OASIS coupler (Craig et al., 2017). The component models within HadGEM3 are the Met Office Unified Model atmospheric model (UM) (Williams et al., 2018) containing the UKCA stratospheric-tropospheric chemistry (Archibald et al., 2020) and GLOMAP-mode aerosol (Mulcahy et al, 2020) schemes with earth surface model JULES (Walters et al., 2017) and the NEMO ocean and CICE sea-ice models (Storkey et al., 2018; Ridley et al, 2018).  In UKESM1 these components are used for interactive simulation of earth system processes, such as the full atmospheric chemistry from UKCA.  The modeling of vegetation and surface properties is particularly important for dust.  In UKESM1 the TRIFFID scheme within JULES simulates interactive vegetation, whilst in HadGEM3-GC3.1 data from the IGBP climatology (IGBP, 2000) is used.  The soil parameters also differ: those in HadGEM3-GC3.1 are based on Van Genuchten (Loveland et al., 2000), whilst those in UKESM1 are from Brooks and Corey (1964).  A full description of UKESM1 is available in Sellar et al. (2019).

The mineral dust is simulated within HadGEM3 by the fully interactive dust scheme described below, which is called each atmospheric model timestep.  The driving fields are calculated directly by the UM and JULES, and dust impacts the rest of the model through radiative interactions with the UM atmosphere and through input to the ocean biogeochemistry in MEDUSA.  In the current configuration the dust is externally mixed with other aerosols.

## 3 Dust Scheme

### 3.1 Description of Dust Scheme

The dust scheme is a development of that described in Woodward (2001) with significant improvements to the emission scheme and newer refractive index data.  Dust emission in six size bins with boundaries at 0.06324, 0.2, 0.6324, 2.0, 6.324, 20.0 and 63.24 µm diameter is calculated at each atmospheric model timestep (20 mins).  Within each bin $dV/d(\log(r))$ is assumed constant, where V is particle volume and r is particle radius, giving a sectional distribution of $dV/d(\log(r))$. Horizontal flux is calculated over a wider size range, with three additional bins with boundaries at 63.24, 200.0, 632.4 and 2000.0 µm diameter.

Flux calculations are based on the method of Marticorena and Bergametti (1995).  Horizonal flux in bin i is given by:

$$G_i = \rho\, B\, U^{*3}\, (1+U^*_{ti}/U^*)\, (1-(U^*_{ti}/U^*)^2)\, M_i\, C\, D\, /\, g \quad (1)$$

where $\rho$ is surface air density in kg m$^{-3}$, B is bare soil fraction, $U^*_{ti}$ is threshold friction velocity for the bin in m s$^{-1}$, $U^*$ is friction velocity excluding orographic effects in m s$^{-1}$, C is a constant of proportionality set to 2.61 from wind-tunnel experiments, D is a dimensionless tuneable parameter (see Section 3.3) and g is acceleration due to gravity in m s$^{-2}$.  $M_i$ is the mass fraction of particles in the bin, obtained from soil clay, silt and sand fractions from HWSD data (Nachtergaele et al., 2008) according to the method described in Woodward (2001).

In UKESM1 dust is emitted only from the bare soil fraction of a gridbox, though there is also an option in the code to allow emissions from seasonally bare sources, based on the leaf area index (LAI).  Dust emissions are prevented if snow is present,

if the ground is frozen, on steep slopes, if soil moisture exceeds a threshold (see below) and at coastal points where the lowest level windspeed over land may be anomalously high. No preferential source terms are used.

The driving fields for the scheme are the model's grid-box mean, time-step mean fields, but equation (1) was derived from shorter timescale measurements at single locations. Corrections are therefore needed to account for the effect of spatial and temporal averaging. Here, model friction velocity $U^*_M$ (in m s$^{-1}$) is multiplied by a dimensionless tuneable constant $k_1$.

$U^* = k_1 U^*_M$   (2)

The value of $k_1$ was chosen empirically, as described in Section 3.3. Ideally such a correction would be spatially and temporally variable to account for rapid fluctuations which might be specific to certain conditions; for example Lunt and Valdes (2002) and Cakmur et al. (2004) related gustiness to surface sensible heat flux, which is typically strongest at midday and within arid regions. These studies suggest that introducing a representation of gustiness can have a large impact and can improve dust simulations, though the magnitude and spatial distribution of the effect appears to be strongly dependent on the parametrization used. Cakmur et al (2004) showed that a parametrization using a linear combination of the gustiness due to dry and moist convection and turbulent kinetic energy produced considerable improvements in modelled optical depth in many areas. The use of a global tuning term rather than a realistic representation of gustiness will introduce biases to our simulations, with friction velocity over-estimated in some areas at some times, and under-estimated in others. Though the higher resolution of our models compared to those of Lunt and Valdes (2002) and Cakmur et al. (2004) (1.875°x1.25° compared with 3.75°x2.5° and 4°x5° respectively), together with the shorter timestep (20 mins compared with 30 minutes interpolated from 6-hourly input and 1 hour respectively) should allow a somewhat better representation of the smaller scale and more variable processes, many phenomena important for dust generation are still sub-grid scale.

Dry threshold friction velocity ($U^*_{td}$) values were obtained from Bagnold (1941). The values for each of the 9 horizontal flux bins are 0.85, 0.72, 0.59, 0.46, 0.33, 0.16, 0.14, 0.18 and 0.28 m s$^{-1}$. The effect of soil moisture on friction velocity is represented using the method of Fécan et al. (1999), which has been shown to agree well with measurements. Threshold friction velocity for moist soil is related to the dry threshold friction velocity by:

$U^*_t / U^*_{td} = 1$   for w<w'

$U^*_t / U^*_{td} = (1 + 1.21(w-w')^{0.68})^{0.5}$   for w>w'   (3)

where w'=$0.14 F_C^2 + 17.0 F_C$

$F_C$ is clay fraction and w is volumetric soil moisture. The model provides average soil moisture over the 10 cm deep top soil level ($w_1$). In order to obtain soil moisture near the surface, as well as to correct for the effects of temporal and spatial averaging, the model soil moisture is multiplied by a dimensionless tunable constant $k_2$, which was also set empirically (see section 3.3):

$w = k_2 w_1$   (4)

The highest clay fraction reported in the measurements on which the algorithm was based was 0.2 (Gillette, 1979). However, it subsequently became clear that the single measurement with this high clay fraction was contaminated by upstream dust (Gillette, pers. Comm., reported in Alfaro and Gomes, 2001). The next highest clay fraction measured was

0.1. As high clay fractions result in unrealistically high emissions from the dust scheme, a maximum of 0.1 is applied, with higher values of $F_C$ being reset to this.

The Fécan et al. treatment of soil moisture was designed for use in arid or semi-arid areas, and in order to apply it to the whole earth, a further constraint on dust production from moist soil is required. Dust production is inhibited when soil moisture exceeds a particle-size dependent threshold ($w_t$), which was chosen to correspond approximately to the maximum soil moisture for which movement was detected in the observations used by Fécan et al..

$$w_t = ( F_C + 0.12 ) / 0.03 \quad (5)$$

The vertical dust flux is calculated for the six emission bins in the range 0.06324 to 63.24 μm, corresponding to the smallest six horizonal flux bins. Total vertical flux relates to total horizontal flux summed across all bins according to the method of Marticorena and Bergametti (1995), which is based on the measurement data of Gillette (1979). The size distribution across the six vertical flux bins follows that of the equivalent horizontal flux bins. Vertical flux in bin i ($F_i$), for i=1 to 6 is given by:

$$F_i = 10^{(13.4 F_c - 6.0)} \; G_i \; \Sigma_{i=1,9} (G_i) / \Sigma_{i=1,6} (G_i) \quad (6)$$

Dust is transported as 6 independent tracers corresponding to the 6 vertical flux bins. Deposition through below-cloud scavenging, turbulent mixing and sedimentation is included, as described in Woodward (2001). The radiative effects of dust on the atmosphere are simulated with the model's 2–stream radiation code (Edwards and Slingo, 1996). Dust radiative properties were derived from the data of Balkanski (2007) for Saharan dust. This is intended to provide the optimum simulation of dust from the Sahara, the world's largest dust source, though the radiative effects of dust from other sources with other mineralogical contents will inevitably be less well modelled (Sokolik and Toon, 1999). No chemical processing of dust in the atmosphere is represented: this is something we hope to include in future versions of the scheme. In UKESM1 the total marine dust deposition flux is passed into the ocean, where it is multiplied by a tuned constant to derive "bioavailable iron" (Yool et al., 2021) which is used by the MEDUSA ocean biogeochemical scheme (Yool et al., 2013) as a nutrient source for plankton growth.

The scheme described here uses the same code as the UM global atmosphere GA7.1 and global coupled GC3.1 configurations, but with two changes in the settings: 1) the three tuning terms (D, $k_1$ and $k_2$) are different and 2) the emission of dust from seasonally vegetated sources which is allowed in GA7 and GC3.1 is deactivated in UKESM1. The latter was done both because it was unclear whether the simulation of the global distribution of the various plant types would be sufficiently accurate for the purpose, and also because the JULES land surface tiling on which the seasonal source code depends had been changed for UKESM1, rendering the associated dust settings invalid. Seasonal sources (see Fig. 10.) accounted for less than 10% of the load in HadGEM3-GC3.1, so this was not expected to have a large impact. The uncertainties associated with the emission scheme are considered in Section 7 .

## 3.2 Diagnostics

The model provides a self-consistent set of diagnostics of dust emissions, depositions and mixing ratios (concentrations). Whilst the concentration, wet deposition, and dry deposition over water are directly comparable to results from other models and to observations, the emission and dry deposition in source regions are not. Atmospheric lifetime, being derived from these terms, is also not comparable. The emission and dry deposition diagnostics include all particles released from the surface, whether or not they are added to the atmospheric load or interact with the model atmosphere in any way. In the Unified Model the flux of particles released from the surface is calculated first and then another section of code performs both the mixing into the atmosphere and the dry deposition back to the surface in a single loop. As a result, a fraction of the particles released from the surface is immediately dry-deposited and never added to the atmospheric burden. In physical terms the re-deposited fraction may be considered as those particles which fall back to the surface within the model timestep. The magnitude of this effect is greater for larger particles due to the size-dependence of sedimentation; and because the particle size range extends to 63 μm it has a considerable impact on the total emission and deposition diagnostics, and also the calculated lifetime.

## 3.3 Dust Tuning

The parameters D, k1 and k2 were tuned to improve agreement between the UKESM1 dust simulation and various observations. The final values chosen were D=$1.0\times10^{-3}$, k1=1.1, k2=0.8 . Sets of UKESM1 experiments were run with different values of these parameters, and the simulations compared with multi-annual means of dust concentration measurements from selected stations of the University of Miami aerosol network (eg Prospero et al., 2010) and AOD measurements from selected AERONET sites (Holben et al., 2001) as well as size distributions from the FENNEC campaign (Ryder et al., 2013). In addition, the model AOD was compared with MODIS data (Levy et al., 2000) and the dust deposition with measurements collected by Huneeus et al. (2011). These observational datasets were also used for evaluation of the final UKESM1 and HadGEM3-GC3.1 dust simulations, and further information is given in Section 5. Note that as the fully spun-up UKESM1 present-day simulation had a slightly different climate from the versions of the model used for tuning, as described below, the results presented in Section 5 do not represent the results of the tuning experiments. The aim of the tuning process was to obtain the optimum simulation of emissions by correcting for terms and processes not included in the climate model but which impact dust emissions, such as the effect of gustiness, the relation between moisture in the model's top soil level and at the soil surface, etc.. In the absence of emission measurements, observations of other dust properties at some distance from the sources must be used for model evaluation. These are affected not only by the emissions but also by transport and deposition processes, and in some cases by the dust radiative properties. Thus the tuning is inevitably influenced by, and compensates for, any biases in these processes and properties in the dust scheme as well as by biases in the driving model fields, such as in bare soil fraction, which affected the dust. In an attempt to reduce the effects of these biases higher importance was given to comparisons with direct observations of dust concentration and size

than with observations of properties such as AOD, and with observations nearer sources than with observations at remote locations. Correct simulation of near-source concentration and size distribution is also important because the impact on the dust radiative forcing and potential feedbacks will be largest in source areas.

Initial sets of tuning experiments were used to explore the parameter space, finding how the model responded to changes in each of the variables and constraining their range; later experiments focussed on finer tuning of the variables. Due to time constraints in the development of UKESM1, dust scheme tuning was carried out in parallel with the later stages of ESM model development. As a consequence of this tuning terms had to be adjusted on an ongoing basis in response to changes in the driving model and there was insufficient time for a highly detailed tuning process. Though otherwise not ideal, this parallel development had the advantage that minor changes could be made to other parts of the ESM to improve the dust simulation. In particular, the lai_min term for grasses in JULES was reduced to 0.3, allowing the grasses to spread more readily and improving the simulation of bare soil fraction (Sellar et al., 2019).

The tuning of UKESM1 dust was carried out primarily with UKESM1-CN, the off-line chemistry version of UKESM1 (Sellar et al., 2019) as this ran more quickly than the full ESM, and the interactive chemistry was expected to have little impact on the dust. Comparisons of dust simulations from the two versions of the ESM showed only small differences, with the various tuning settings tending to perform similarly in each case. Pre-industrial (PI) simulations were used for the tuning as a fully spun-up present-day experiment was not available at that stage. Whilst this meant that the model climate did not match the period of the observations, this discrepancy was likely to have had a relatively small effect compared with the cumulative effect of the many uncertainties in the dust scheme, such as biases in the simulated bare soil fraction, soil moisture and wind-speed, resolution-related limitations in capturing some emissions mechanisms, and the omission of surface-crusting effects and other poorly understood processes. Tuning tests were also carried out with "pseudo-present-day" experiments, initialised from PI runs, but then run to equilibrium with present-day (PD) atmosphere and land-use settings. These produced similar results to the main tuning experiments and gave confidence in the use of PI simulations to establish tuning parameters. However, the climate in the fully spun-up present-day UKESM1 simulation reported here was slightly different from that in the tuning runs.

The dust in HadGEM3-GC3.1 was not re-tuned for this work. HadGEM3 is a widely used model (Williams et al., 2018); it has been regularly updated, with some dust re-tuning being carried out before release of a new version if the update had a major effect on the dust. Here we make use of the existing HadGEM3-GC3.1 configuration (Kuhlbrodt et al., 2018) as used for the CMIP6 simulations (Eyring et al., 2016), without any changes, except in experiments H3_TUK_EXSS and H3_TUK_INSS, as described in Section 4. The dust tuning parameters had been set to $D=2.25\times10\text{-}4$, $k_1=1.45$, $k_2=0.5$ .

## 4 Experiments

The results presented here were primarily obtained from UKESM1 and HadGEM3-GC3.1 historical experiments parallel to those performed as part of CMIP6 (Eyring et al., 2016), but with extra diagnostics. In order to obtain diagnostics needed for

calculating DRE, a copy of each relevant CMIP6 experiment was run for 20 years with the "double call" method. This involved calling the radiation scheme twice each timestep, with the radiative effect of dust excluded from the first call but included in the second which is used to progress the model. The dust DREs are then calculated as the differences between the fluxes from each call. In addition, two parallel HadGEM3-GC3.1 experiments were run: in both of these the tuning terms were set to UKESM1 values, and in one the seasonal sources were also deactivated. These allowed us to investigate the relative importance of the various differences between the main simulations. Two UKESM1 AMIP experiments from CMIP6 were also used. The experiments are summarised in Table 1.

| Name | Model | Parallel CMIP6 Experiment | Meaning period | Tuning Settings $D,k_1,k_2$ | Seasonal Sources |
|---|---|---|---|---|---|
| UK_PI | UKESM1 | piControl | 20 yrs | (UKESM1) | (off) |
| H3_PI | HadGEM3-GC3.1 | piControl | 20 yrs | (HadGEM3_GC3.1) | (on) |
| H3_PD | HadGEM3-GC3.1 | Historical | 1995-2014 | (HadGEM3_GC3.1) | (on) |
| UK_PD | UKESM1 | Historical | 1995-2014 | (UKESM1) | (off) |
| H3_TUK_EXSS | HadGEM3-GC3.1 | Historical | 1995-2014 | UKESM1 | off |
| H3_TUK_INSS | HadGEM3-GC3.1 | Historical | 1995-2014 | UKESM1 | (on) |
| UK_S5 | UKESM1 | SSP5-8.5 | 2081-2100 | (UKESM1) | (off) |
| H3_S5 | HadGEM3-GC3.1 | SSP5-8.5 | 2081-2100 | (HadGEM3_GC3.1) | (on) |
| UK_S2 | UKESM1 | SSP2-4.5 | 2081-2100 | (UKESM1) | (off) |
| H3_S2 | HadGEM3-GC3.1 | SSP2-4.5 | 2081-2100 | (HadGEM3_GC3.1) | (on) |
| A_UK_PI | HadGEM3-GA7.0 | piClim-control | 30 yrs | UKESM1 | off |
| A_UK_PI_V2014NOLU | HadGEM3-GA7.0 | piClim-histNoLU | 30 yrs | UKESM1 | off |

**Table 1: Summary of experiments. The UKESM1 tuning settings for D,k_1 and k_2 are 1.0x10⁻³, 1.1 and 0.8 respectively; the HadGEM3 settings are 2.25x10⁻⁴,1.45,1.1. Tuning Settings and Seasonal Source settings in brackets indicate that the values are the usual ones for the model used in that experiment. For further explanation of these terms see Section 3.**

## 5 Present-day dust simulations and evaluation

### 5.1 Concentration and load

Figure 1 shows dust load from the UKESM1 and HadGEM3-GC3.1 present-day simulations (UK_PD and H3_PD), together with the differences and fractional differences. Both models capture the expected global dust distributions qualitatively well. The UK_PD global load of 19.5 Tg is 30% higher than the 15.0 Tg H3_PD value. These values span the AeroCom phase III mean of 16.6 Tg, and are within the range of 5.7 Tg to 22.3 Tg shown by participating models (Gliss et al., 2021). They are also consistent with the more observationally constrained estimate of Kok et al. (2017) for PM20 dust load of 23 Tg with a range from 14 Tg to 33 Tg. The greatest differences between the models are over the Sahel, India, the Middle East, Asian midlatitudes and Australia, and are predominantly due to differences in the bare soil and associated changes (see Section 5.6 and Fig. 8).

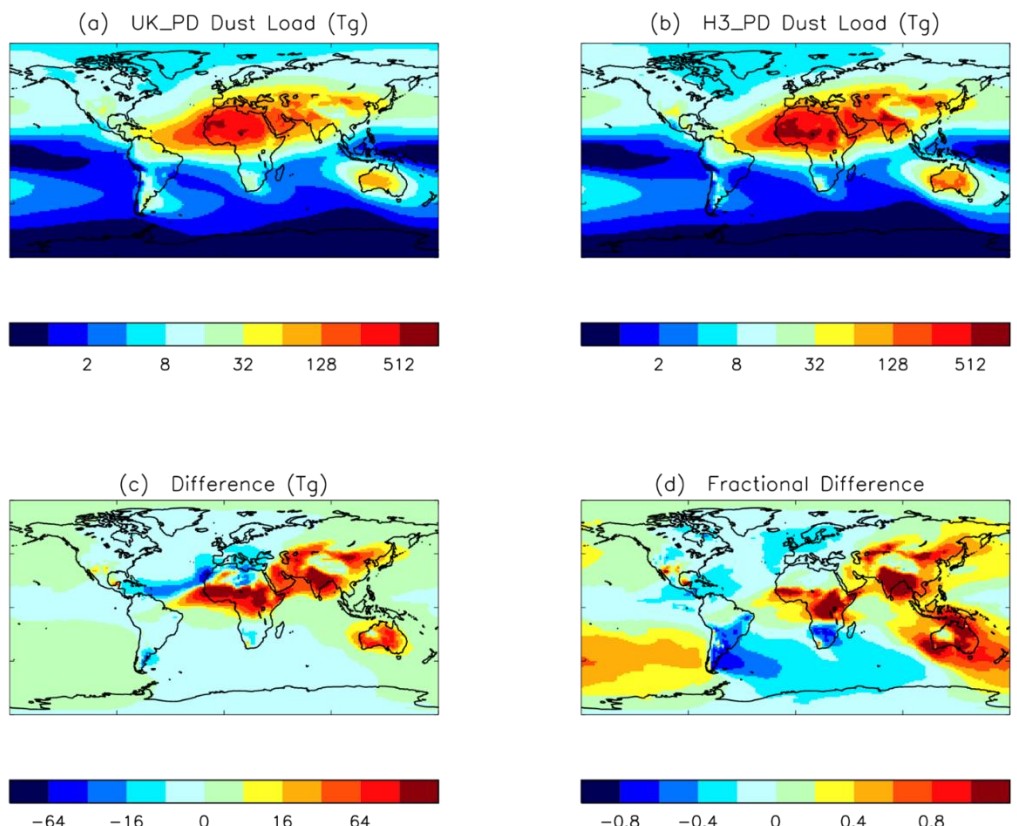

**Figure 1: Dust load (Tg) from H3_PD and UK_PD, difference and fractional difference.**


Figure 2 shows scatterplots of seasonal mean modelled dust concentrations versus observations from remote sites from the University of Miami network and AERONET sites (Holben et al., 2001) chosen for data availability and dustiness based on the Angstrom exponent (see Section 5.3). At least four years of monthly data was available for the chosen sites. Both UK_PD and H3_PD show good agreement with the observations, with the models being within a factor of three of the

observations at most stations for most seasons. The correlation coefficients for the concentration and AOD data shown in Fig. 2 are 0.89 for UKESM1 and 0.87 for HadGEM3-GC3.1. The slight high bias at a few North Pacific stations in spring and summer is increased in UKESM1 due to larger bare soil areas in Asia simulated by TRIFFID, with associated increased windspeed and reduced soil moisture. Similarly, increased bare soil in Australia results in an increased high bias at the nearest South Pacific station. The low biases almost all occur at two stations on the Antarctic Peninsula, where the dust from

Patagonia dominates. The concentrations here are likely to be very sensitive to the westerly winds in that region. The bias is worsened in UKESM1 due to a low bias in the simulated bare soil fraction. Overall, the UKESM1 surface concentrations

show good agreement with observations, compared with other ESMs (Checa-Garcia et al, 2021). The similarity in the level of performance between UKESM1 and HadGEM3-GC3.1 is noteworthy, given the many extra processes and feedbacks within the ESM, and in particular its use of interactive vegetation.


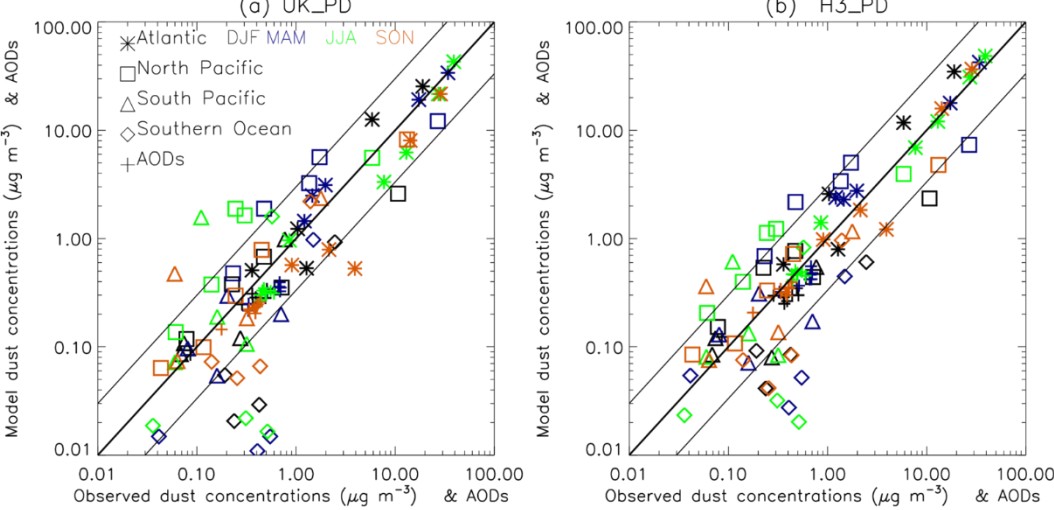

**Figure 2: Scatterplots of model versus observational multi-annual seasonal mean concentrations at remote marine locations and AODs in dusty areas for (a) UK_PD and (b) H3_PD. Concentrations are from the University of Miami Network stations at Cape Grim, King George Island, Mawson Station, Palmer Station, Funafuti, Nauru, Norfolk Island, American Samoa, Midway, Oahu,**
**Cheju, Fanning, Enewetak, Barbados, Bermuda, Mace Head, Miami, and Izana. AODs are from AERONET vn3 at the stations listed in Fig. 4.**

## 5.2 Size distribution

UKESM1 dust was tuned to give better agreement with the size distribution data from the FENNEC campaign (Ryder et al.,
2013) than was shown in HadGEM3-GC3.1. This data was particularly useful as it provided measurements near sources where the effects of deposition and transport on the size distribution would be relatively small, and, unusually, it included data from several instruments for measuring larger particles with diameters above 3μm. The FENNEC campaign took place in June 2011 in the remote Sahara.

Figure 3 presents normalised volume size distributions from 20 year June means from UK_PD and H3_PD compared with a
fit to the FENNEC observations. UK_PD shows good agreement with the observations throughout the size range. The lower level data has a somewhat greater coarse particle fraction than the higher level data, as might be expected in a source region, given the short lifetimes of the largest particles. The peak diameter is slightly smaller than in the observations. This would be consistent with the 20 year June mean containing a smaller fraction of freshly emitted particles than were measured in observations which were intended to sample dusty conditions.

H3_PD concentrations overlap with the FENNEC observations, though the overall agreement is poorer as the model is outside the range of the observations for the two smallest bins and the peak of the distribution is in bin 4 (2–6 μm diameter), rather below the peak diameter of the observations.

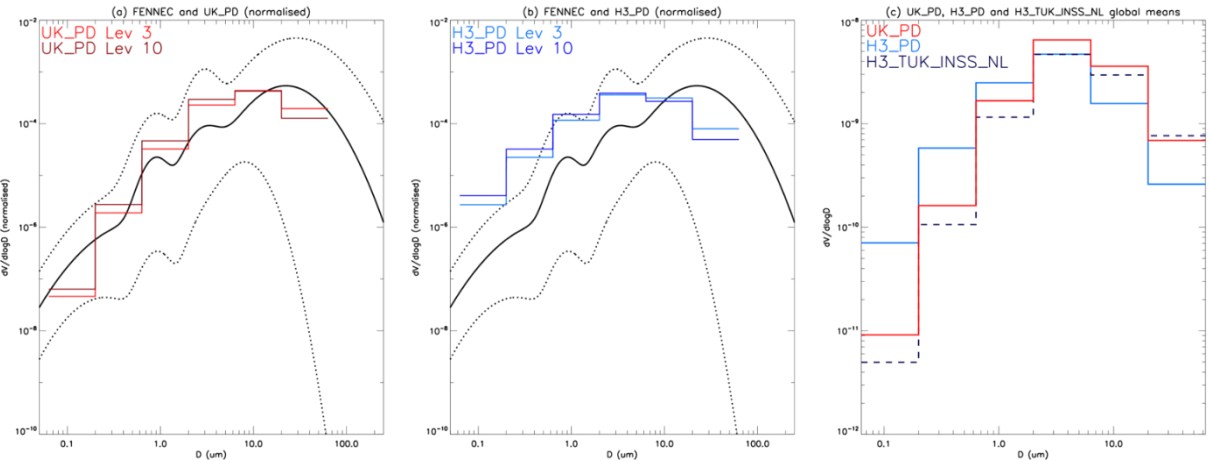


**Figure 3: Dust volume size distributions: (a) normalised distribution from a fit to FENNEC mean, maximum and minimum data (Ryder,2013) and from a multi-annual June mean of UK_PD data meaned over a rectangular area 13W-4W, 21S-26N covering the FENNEC campaign region, for model levels 3 (approx. 96m) and 10 (approx. 770m), corresponding to the height range of most of the FENNEC measurements; (b) as (a) but for H3_PD; (c) global mean size distribution from UK_PD, H3_PD and**
**H3_TUK_INSS_NL.**

### 5.3 Optical depth

Comparison of monthly mean model AODs with data from AERONET (Holben et al.,2001) sites in dusty areas (Fig. 4) shows that the modest low bias in H3_PD is slightly worsened in UK_PD, though the model mean is within 2 standard
deviations of the observations in almost all cases. The stations were selected to obtain as realistic observational climate means as possible given the considerable variability of dust and the short data record, and to minimise the effects of other aerosols. Sites were chosen from those in potentially dusty areas, as having a minimum of 4 years of monthly data with at least 10 daily means per month, and a monthly mean Angstrom exponent (870 - 440) below 0.5 for at least 10 months of the year. The only sites to fulfil these criteria were: Tamanrasset_INM in the Sahara, Capo Verde off the west African coast and
six Sahelian stations all in a narrow band between 12° and 16° north.

The observed multi-annual mean AOD at 440 nm meaned over these AERONET sites is 0.45, whilst UK_PD simulates 0.28 and H3_PD simulates 0.37. Comparison of the simulations shows that the lower AOD in UK_PD is due to a combination of lower dust optical depth (DOD), (0.13 in UK_PD and 0.19 in H3_PD) with lower optical depth due to other species (0.15 in UK_PD and 0.18 in H3_PD). The lower DOD in UK_PD is caused by the larger particle sizes in that model, as evidenced

by the Angstrom exponent for the DOD (440-870) which is -0.06 in UK_PD and 0.23 in H3_PD, whilst the mean atmospheric load at these sites is higher in UK_PD: 540 mg m$^{-2}$, compared with 340 mg m$^{-2}$ in H3_PD. The Angstrom exponent for the total AOD (440-870) is 0.65 in UK_PD and 0.76 in H3_PD, compared with 0.33 in the AERONET observations. This indicates too low a coarse mode fraction in the simulations which suggests too little dust, as this is the dominant coarse mode species at these locations and the low simulated AODs show that there is no excess of fine mode

aerosol. The slightly better agreement of UK_PD with the observed Angstrom exponent may be an indication that the particle size distribution is more realistic than in H3_PD, which would be consistent with the comparisons with FENNEC data, though with the caveat that the potential effects of different concentrations of other aerosol species at the AERONET sites cannot be ignored.

Studies by Marticorena et al. (2010, 2017) have shown that the annual cycle of dust in the western Sahel is related to the

timing of the West African Monsoon (WAM) and the annual north-south shift in the ITCZ. Through winter and early spring the Harmattan flow brings dust from the Sahara to this region, and the increasing AODs over the first few months of the year are consistent with the annual cycle of Saharan dust, as seen at Tamanrasset-INM. From May to October the WAM dominates the region, bringing precipitation and winds from the southwest. During the early part of the monsoon season strong but sporadic local emission events driven by Mesoscale Convective Systems (MCS) (Caton-Harrison et al., 2019 and

2020) produce similar levels of dust to those seen in the dry season. The UK_PD simulations slightly underestimate the AODs due to Saharan dust in the dry season, but the main sources of error appear to be in the timing of the arrival of the WAM, with AODs already decreasing in April, and in particular a failure to simulate the dust production from local sources. These failings are unsurprising, given the difficulties in simulating the WAM and MCS in an N96 climate model with a resolution of 1.85° x 1.25° (Marsham et al.,2011; Heinold et al., 2013), and in simulating realistic vegetation cover in this

marginal area. In this regard, the performance at Sahelian sites cannot be considered representative of the dust simulation generally.At Tammanraset UKESM1 simulates the dry season dust well, but underestimates the wet season AOD. Similar patterns are seen in the AOD at Capo Verde which is dominated by Saharan dust, though sea salt is also present at this site (Fomba et al., 2014, Ryder et al., 2018). Guiardo et al (2014) identify four source areas for Tamanrasset: dust from an area immediately south of the site and from the east Libyan desert affects the site all year, whilst dust from the western Sahara

and the Libya-Tunis border only reach Tamanrasset in the wet season. Some of these areas could be affected by errors in the simulation of the WAM and the position of the ITCZ; windspeed bias could also be involved, as could resolution-related issues such as difficulties in the simulation of Low Level Jets or the representation of local orography, as well as biases in the dust scheme.


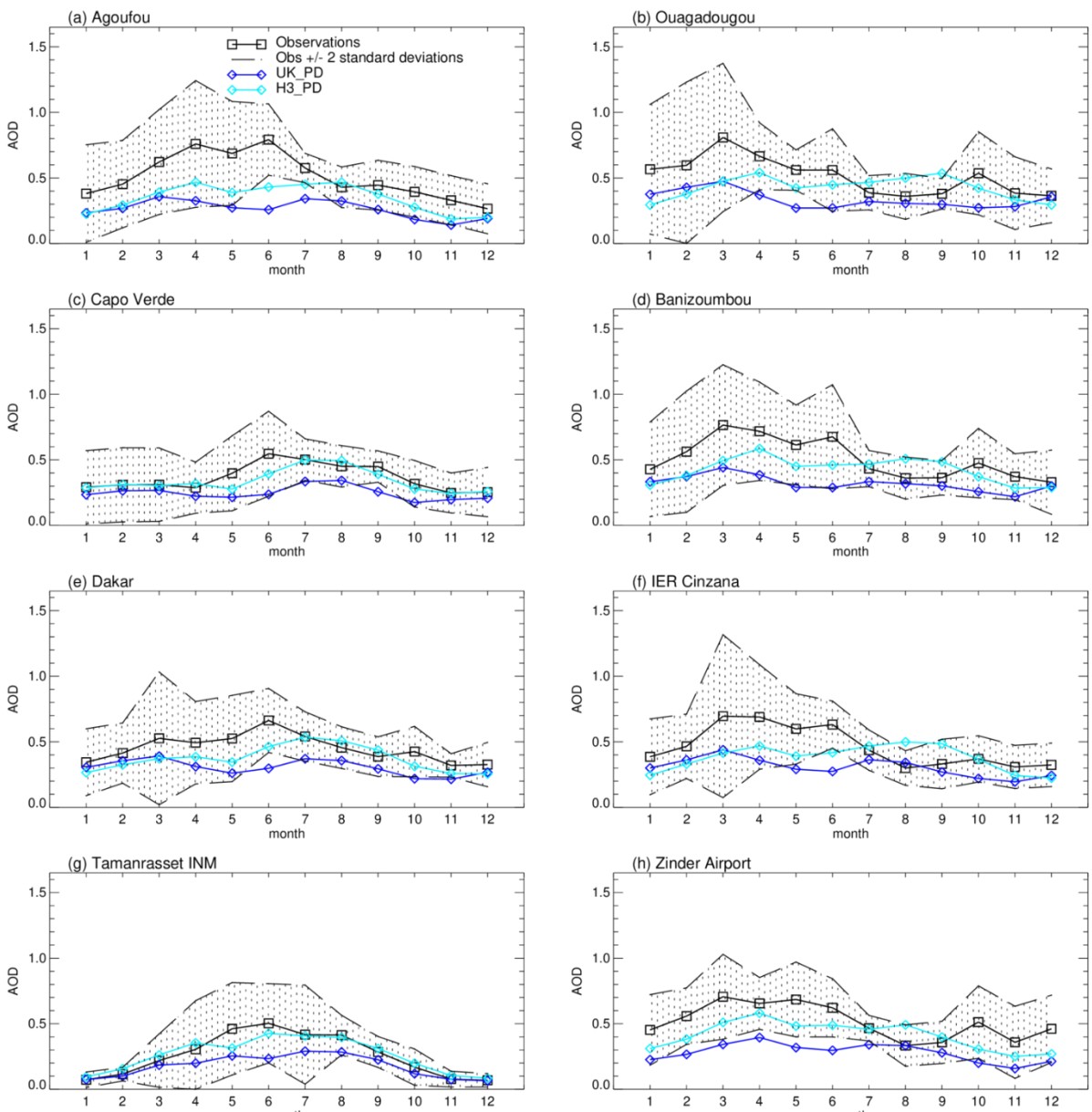

**Figure 4: Annual cycle of AOD (440nm) at dust-dominated AERONET stations.from UK_PD and H3_PD. Site locations: Agoufou [1.48W,15.35N], Oagadougou [1.40W, 12.20N], Capo Verde [22.94W,16.73N], Banizoumbou [2.66E, 13.54N], Dakar [19.96W, 14.39N], IER Cinzana [5.93W, 13.28N], Tamanrasset INM [5.53E, 22.79N], Zinder airport [8.99E, 13.78N].**


Figure 5, which compares the total aerosol optical depth at 550nm in UK_PD with MODIS data shows a low bias in the Sahel as mentioned above, and also in dust from the Bodele depression. The Bodele dust source is very difficult to simulate in global climate models because a resolution of a few tens of km is needed to represent the Bodele Low Level Jet, which is

responsible for much of the dust emissions (Todd et al., 2007). The dust-dominated AOD over the northern Sahara and the Arabian Peninsula is also low, whilst over Australia it is too high. The high bias on the southwest side of the Himalayas and low bias on the northeast side suggest the model may be failing to transport aerosol over the steep orography there, or may be associated with other aerosol species. These biases reflect model weaknesses and also the difficulty of finding a single set of tuning terms which gives a good performance against all metrics: settings which gave improvement in AOD over North Africa resulted in worsening of biases associated with Asian and Australian dust.

The effects of non-sphericity on dust particle optical properties were not included in the models. It has been estimated that the extinction efficiency could increase by as much as 29% (Kok et al., 2017). An indication of the potential effect this might have was estimated by multiplying the UK_PD dust optical depth by a factor of 1.29, which resulted in improved agreement with the satellite data (Fig. 5).

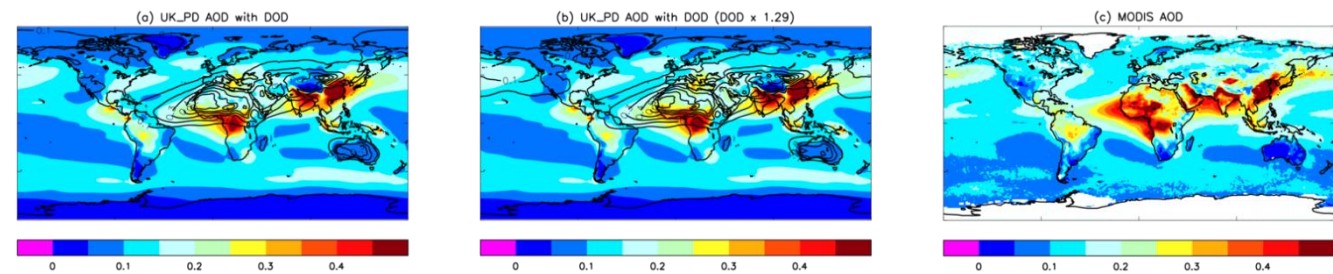

**Figure 5**: **AOD (550nm) from (a) UK_PD, with contours showing the fractional contribution of dust to total AOD; (b) as (a) but with dust optical depth multiplied by 1.29; (c) MODIS 2003-2012 mean.**

## 5.4 Deposition

Though the model deposition diagnostics will not represent observed deposition in source regions (see above), they are comparable to measurements in locations where there are no local emissions. Deposition rates have been evaluated against annual mean measured fluxes from a range of data sources at sites remote from emission areas, as selected by Huneeus et al. (2011) for the evaluation of dust in AeroCom models. These include deposition observations from Ginoux at al. (2001), dust and iron deposition fluxes reported in Mahowald et al. (2009), ice core data from Mahowald et al (1999) and sediment trap data from the DIRTMAP database (Tegen et al., 2002; Kohfeld and Harrison, 2001). Though this collection of deposition data from 84 sites is one of the most comprehensive available, there are considerable uncertainties associated with the observations: most notably that, although sites with less than 50 days of data were excluded, some of the records are not long enough to be considered climatological (Huneeus, 2011). This is particularly problematic for dust fields which show very strong variability: a large fraction of annual deposition may occur over a few days of the year (Prospero et al.,2010).

A scatterplot of UK_PD mean deposition rates against these observation (Fig. 6) shows that the model agrees reasonably well with the observations, and model results are within a factor of 10 of the observations at most locations. The only area with a noticeable bias is in Antarctica and the Southern Ocean. The four Antarctic stations where the model significantly overestimates deposition rates are very close together near Dumont station, between 64.60–64.97S and 141.07–141.45E. This localised bias may be due to overestimated windspeed or to the underestimated sea ice cover which has been observed

in UKESM1 (Sellar et al., 2019) and which would result in increased roughness and thus reduced aerodynamic resistance, leading to increased deposition velocity (Woodward, 2001) Superficially, the results appear broadly comparable to those of the AeroCom models reported in Huneeus et al.(2010), though those data represent the output of models run for a single year, in most cases with winds derived from re-analyses. UKESM1 deposition rates also compare well to those of other ESMs in the CRESCENDO project, as recorded by Checa-Garcia et al. (2021).


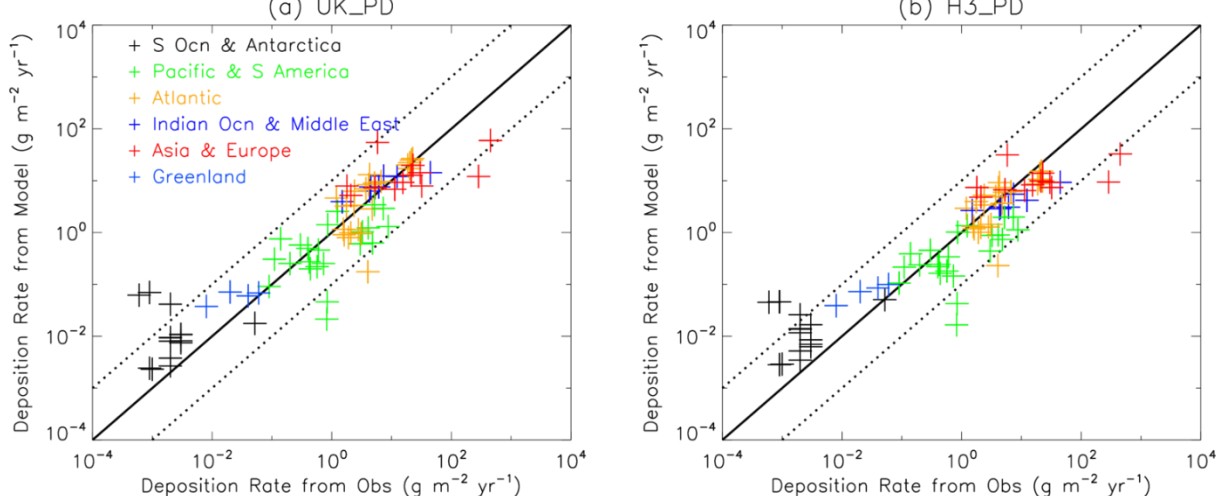

**Figure 6: Scatterplot of deposition rates (g m-2 yr-1) from UK_PD and H3_PD vs observations (Huneeus, 2011)**

### 5.5 Radiative effects

Figure 7 and Table 2 show the dust direct radiative effect (DRE) - the change in flux directly due to the presence of dust, excluding any secondary effects. This is calculated using the "double call" method as described in Section 4.

In UKESM1 dust is generally more reflective than the surface in the shortwave (SW), except over ice and the brightest deserts, with the result that the net downward shortwave at the top of the atmosphere (ToA) is reduced everywhere but over these very light surfaces. The global mean ToA SW dust DRE is -0.280 W m$^{-2}$, and -0.410 W m$^{-2}$ in the clearsky (CS). The

ToA longwave (LW) DRE is positive everywhere, with higher values over areas of higher load, particularly the Sahara. It has a global mean of 0.194 W m$^{-2}$ (0.237 W m$^{-2}$ CS). SW and LW combine to give a positive net DRE over the Sahara with

a maximum of 3.71 Wm$^{-2}$, but partially cancel in most other regions to produce modest positive net values over lighter surfaces and elsewhere negative net values down to -3.05 Wm$^{-2}$ in the Atlantic under the Saharan plume, giving a global mean of only -0.086 W m$^{-2}$.  At the surface the global mean net DRE is -0.168 W m$^{-2}$, being positive over the brightest surfaces and negative elsewhere, with a maximum of 4.45 Wm$^{-2}$ in the Sahara and a minimum of -3.19 Wm$^{-2}$ under the Saharan plume.  The surface SW DRE is negative everywhere and has a global mean of -0.556 W m$^{-2}$ (-0.679 W m$^{-2}$ CS); the LW is positive everywhere, with a global mean of 0.388 W m$^{-2}$ (0.455 W m$^{-2}$ CS).

| Experiment | Load (Tg) | ToA SW (W m$^{-2}$) | ToA LW (W m$^{-2}$) | ToA Net (W m$^{-2}$) | Surf.SW (W m$^{-2}$) | Surf. LW (W m$^{-2}$) | Surf. Net (W m$^{-2}$) |
|---|---|---|---|---|---|---|---|
| H3_PD | 15.01 | -0.460 | +0.164 | -0.296 | -0.688 | +0.338 | -0.350 |
| (1) Difference due to tuning (size distribution) | - | +0.276 [ -60] | -0.021 [ -13] | +0.255 [ -86] | +0.288 [ -42] | -0.026 [ -8] | +0.262 [ -75] |
| H3_TUK_INSS_NL | 15.01 | -0.184 | +0.143 | -0.041 | -0.400 | +0.312 | -0.088 |
| (2) Difference due to tuning (load) | -4.63 [ -31] | +0.057 [ -31] | -0.044 [ -31] | +0.013 [ -31] | +0.123 [ -31] | -0.096 [-31] | +0.027 [ -31] |
| H3_TUK_INSS | 10.38 | -0.127 | +0.099 | -0.028 | -0.277 | +0.216 | -0.061 |
| (3) Difference due to seasonal sources | -0.98 [ -9] | +0.011 [ -9] | -0.009 [ -9] | -0.002 [ -9] | -0.027 [ -10] | -0.020 [ -9] | +0.007 [ -12] |
| H3_TUK_EXSS | 9.40 | -0.116 | +0.090 | -0.026 | -0.249 | +0.196 | -0.053 |
| (4) Difference due to driving model | +10.14 [+108] | -0.164 [+141] | +0.103 [+115] | -0.061 [+236] | -0.307 [+123] | +0.192 [+98] | -0.115 [+215] |
| UK_PD | 19.54 | -0.280 | +0.194 | -0.086 | -0.556 | +0.388 | -0.168 |
| Difference from H3_PD to UK_PD | +4.52 [ +30] | -0.180 [ -39] | -0.030 [ +18] | -0.210 [ -71] | -0.132 [ -19] | -0.049 [+15] | -0.182 [ -52] |

Table 2: Global mean load and all-sky direct radiative effects due to dust in a present-day climate, from various experiments. Alternate rows show experimental results and the absolute differences, followed by percentage differences in brackets, between the experiment below and the experiment above.

In HaGEM3-GC3.1 the global mean net DRE at ToA is -0.269 W m$^{-2}$, varying between -7.18 W m$^{-2}$ and 1.90 W m$^{-2}$; and at
the surface the mean is -0.350 W m$^{-2}$, with a range from -5.75 W m$^{-2}$ to 2.88 W m$^{-2}$.  The larger shortwave and smaller
longwave effect are associated with the difference in size distribution between the UK_PD and H3_PD simulations, as will
be explored in Section 5.6.

The relatively small global mean net DREs are the residuals of the partial cancellation of areas of larger positive and
negative net DREs, as well as the partial cancellation of component DREs of different sign, each of which are sensitive to
changes in the dust load, spatial distribution, size distribution and radiative properties.  As a result, there are large
uncertainties associated with estimates of global mean dust DREs.

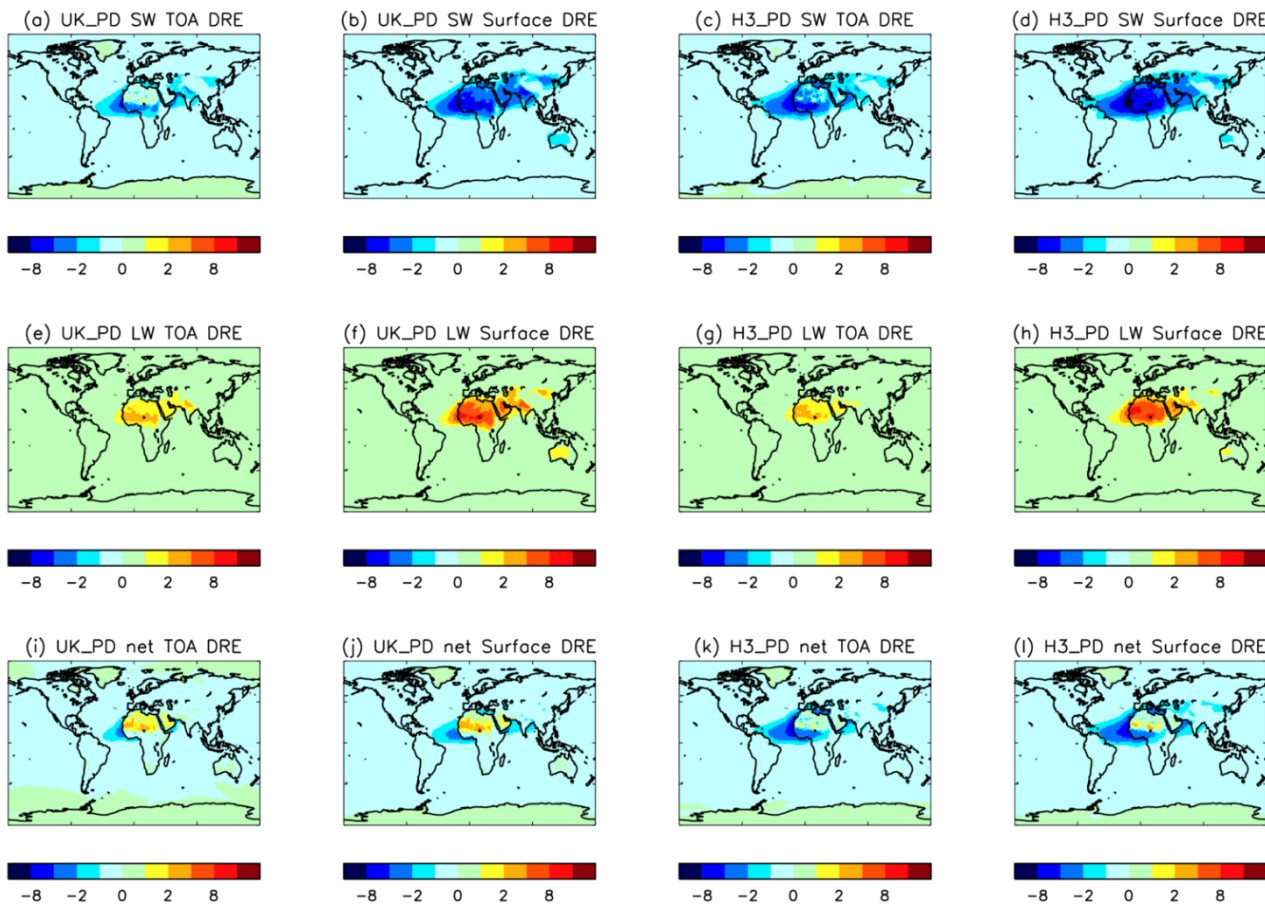

**Figure 7: Dust direct radiative effects (Wm$^{-2}$) in UK_PD and H3_PD.**


## 5.6 Drivers of UKESM1 – HadGEM3-GC3.1 differences

Running simulations with almost identical dust schemes in two similar models allows us to explore the impacts of a number of factors on the dust simulation. The causes of the differences between the HadGEM3-GC3.1 and UKESM1 present-day dust load and DREs may be divided into four groups: 1) The change in size distribution with re-tuning; 2) The change in total load due to re-tuning; 3) The (de-)activation of seasonal sources; and 4) The change of driving model. These are investigated using results from H3_PD, UK_PD and two additional experiments representing intermediate configurations. H3_TUK_EXSS is a HadGEM3-GC3.1 experiment with UKESM1 dust tuning and excluding seasonal sources, to help isolate the effects of the change in model. H3_TUK_INSS is parallel to this but includes seasonal sources to allow investigation of the effect of these sources (see Table 1). Another set of dust DRE results H3_TUK_INSS_NL have been generated by normalising the DREs from H3_TUK_INSS by the ratio of H3_PD load to H3_TUK_INSS load. This allows us to separate the effects of the change in size distribution (from comparing H3_TUK_INSS_NL and H3_PD) from the effects of the load change (from comparing H3_TUK_INSS_NL and H3_TUK_INSS). Global mean volume size distributions from UK_PD, H3_PD and H3_TUK_INSS_NL are shown in Fig. 3c. The implicit assumption that the DREs are proportional to total load for a given size distribution is a reasonable first approximation, although it ignores the feedback between dust DREs and emissions (Miller et al., 2004, Woodage and Woodward, 2014; Kok et al., 2018). Load and DREs from each experiment, together with the differences due to each factor, are listed in Table 2

Overall, the changes from H3_PD to UK_PD result in an increase in total load of 30%. The load change due to re-tuning (Factor 2) is responsible for a reduction of 31%, whilst the disabling of seasonal sources (Factor 3) only decreases global load by a further 9%. The change in driving model (Factor 4) more than doubles the load and produces much the largest individual impact on the atmospheric burden. The most important element in this change is the vegetation: the disparity between the bare soil simulated by TRIFFID in UKESM1 and the IGBP climatology used in HadGEM3-GC3.1 accounts for 70% of the extra emissions in UK_PD compared with H3_TUK_EXSS. The global average bare soil fraction is 0.26 in UK_PD and 0.24 in H3_UK_EXSS, but regional variations are much larger and their geographic distribution promotes extra dust production in UK_PD. The 29% of the UK_PD bare soil area that is vegetated in H3_TUK_EXSS is mostly in arid areas at the margins of existing deserts where conditions favour dust emission, whilst the 25% of the H3_TUK_EXSS bare soil area that is vegetated in UK_PD is mainly in regions where moisture limits dust emission (Fig. 8g). Areas where there is less vegetation and thus reduced roughness might be expected to be associated with higher near-surface wind speed and hence increased evaporation and reduced soil moisture, providing conditions particularly favourable to dust production. This can be seen in the regions of greatest difference between UK_PD and H3_TUK_EXSS such as the Sahel, India and the Kazakh Steppe (Fig. 8g,d,k,o)...

Global mean dust net DREs in UKESM1 are smaller than in HadGEM3_GC3.1, despite the larger load (Table 2, Fig. 7). The net surface DRE of -0.168 W m$^{-2}$ is about half the H3_PD value of -0.350 W m$^{-2}$, and the -0.086 W m$^{-2}$ net ToA DRE is less than one third of the H3_PD value of -0.296 W m$^{-2}$. These global means are the residuals of the partial cancellation of

SW and LW DREs, as well as of spatial meaning of areas of net DRE with different signs. Change in size distribution
consequent on re-tuning (Factor 1) is the factor responsible for the greatest absolute change in the net DREs, due to its large
impact in the SW not being balanced by its much smaller LW effect. SW DREs are approximately halved in response to this
size distribution change, mainly because of reduction in the number of finer particles in bin2 (0.2–0.63 μm) which have a
strong SW effect. In contrast, LW DREs are only reduced by about 10%, chiefly because the alteration in the number of
coarser particles is relatively small. In particular, the global number of bin 4 particles (2.0–6.3 μm), which dominate the LW
effect, is almost unaltered, with increases in the Saharan plume and northern mid-latitudes balanced by reductions elsewhere,
whilst the numbers of bin 5 and 6 particles (6.3–63 μm) which are somewhat less radiatively active are not greatly increased.

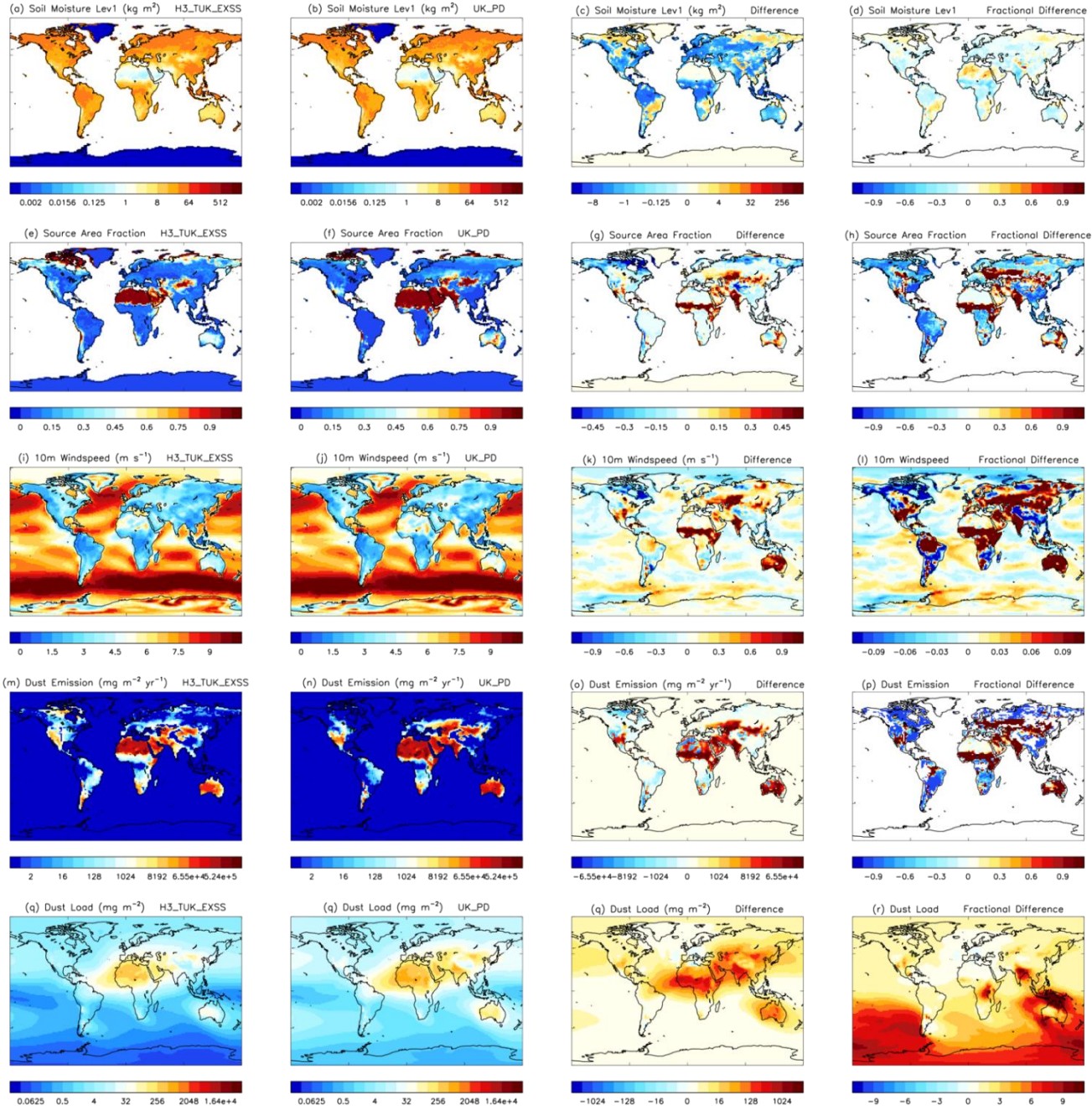

**Figure 8: Soil moisture in top model layer, dust source area fraction, 10m wind-speed, dust emission diagnostic and dust load from H3_TUK_EXSS and UK_PD, and the differences and fractional differences between experiments.**

## 6 The response of dust to changing climate

### 6.1 Pre-industrial to present-day changes

The global mean pre-industrial to present-day dust forcing (defined here as the change in DRE) is small in both models. It is
calculated as the difference between dust DREs in the 20 year mean UK_PD (and H3_PD) runs, and a 20 year mean from
the respective PI control runs. UKESM1 (HadGEM3-GC3.1) simulates a forcing of +0.007 W m$^{-2}$ (-0.029 W m$^{-2}$) at ToA
and +0.005 W m$^{-2}$ (-0.038 W m$^{-2}$) at the surface, from a change in dust load of -0.10 Tg ( +1.40 Tg).

The role of the vegetation response in the historical dust changes is investigated using the results of atmosphere only AMIP
experiments in which the driving fields, except vegetation, were taken from a UKESM1 pre-industrial simulation. In the
control experiment (A_UK_PI) the vegetation fields - LAI, canopy height and vegetation fraction - were also taken from the
pre-industrial simulation; in the parallel A_UK_PI_V2014NOLU experiment the 2014 vegetation values from a UKESM1
historical experiment which excluded anthropogenic land-use changes were used. In both experiments the dust settings were
as in UKESM1. The difference between these simulations gives an estimate of the effect of the change in vegetation alone
on dust. The effect of climate change excluding the vegetation response (but including land-use change) is estimated from
the difference between HadGEM3-GC3.1 simulations H3_PD and H3_PI.

Figure 9 shows the differences in dust load due to vegetation response and to climate estimated in this way, together with the
sum of these two changes, and the equivalent change in UKESM1 from the pre-industrial UK_PI to the present-day UK_PD.
The similarities between the patterns of load change due to the combined vegetation and climate changes and due to the
changes in UKESM1 is notable, giving confidence that dust changes simulated by the models are comparable even though
the present-day dust is somewhat different. From these experiments the global totals of the PD to PI differences in dust load
are -1.04 Tg due to the vegetation response, +1.40 Tg due to climate (and land-use) change and -0.10 Tg due to other model
differences, including the impact of the extra ESM processes and feedbacks included in UKESM1. The difference between
the UKESM1 PI to PD changes and the sum of the vegetation and climate driven changes is caused by a combination of the
interactions of earth system processes, the differences between the dust settings in HadGEM3-GC3.1 and UKESM1, and the
natural variability of dust. In our models, the impact of the vegetation response on dust was comparable in magnitude but
opposite in sign to the direct impact of climate change over the historical period

### 6.2 Present-day to future changes

The dust changes due to possible future changes in climate are explored using a set of scenario experiments representing
future Shared Socio-Economic Pathways (Sellar et al., 2020; Riahi et al., 2017). Means of the last 20 years (2081-2100) of
SSP5-8.5 and SSP2-4.5 experiments UK_S5, H3_S5, UK_S2 and H3_S2 are compared with the present day (1995-2014)
period of the historical experiments UK_PD and H3_PD. The Shared Socio-Economic Pathway SSP5-8.5 represents high-
end projections of fossil fuel and energy use, food demand and greenhouse gas emissions, assuming fossil-fuelled
development; it has a radiative forcing pathway similar to the highest Representative Concentration Pathway RCP8.5 (Moss

et al., 2010; van Vuuren et al., 2011). SSP2-4.5 represents a "Middle of the Road" future, with social, economic and technological developments broadly following historical patterns, giving a radiative forcing pathway similar to RCP4.5. The scenario experiments are initialised from the end of the respective historical runs.

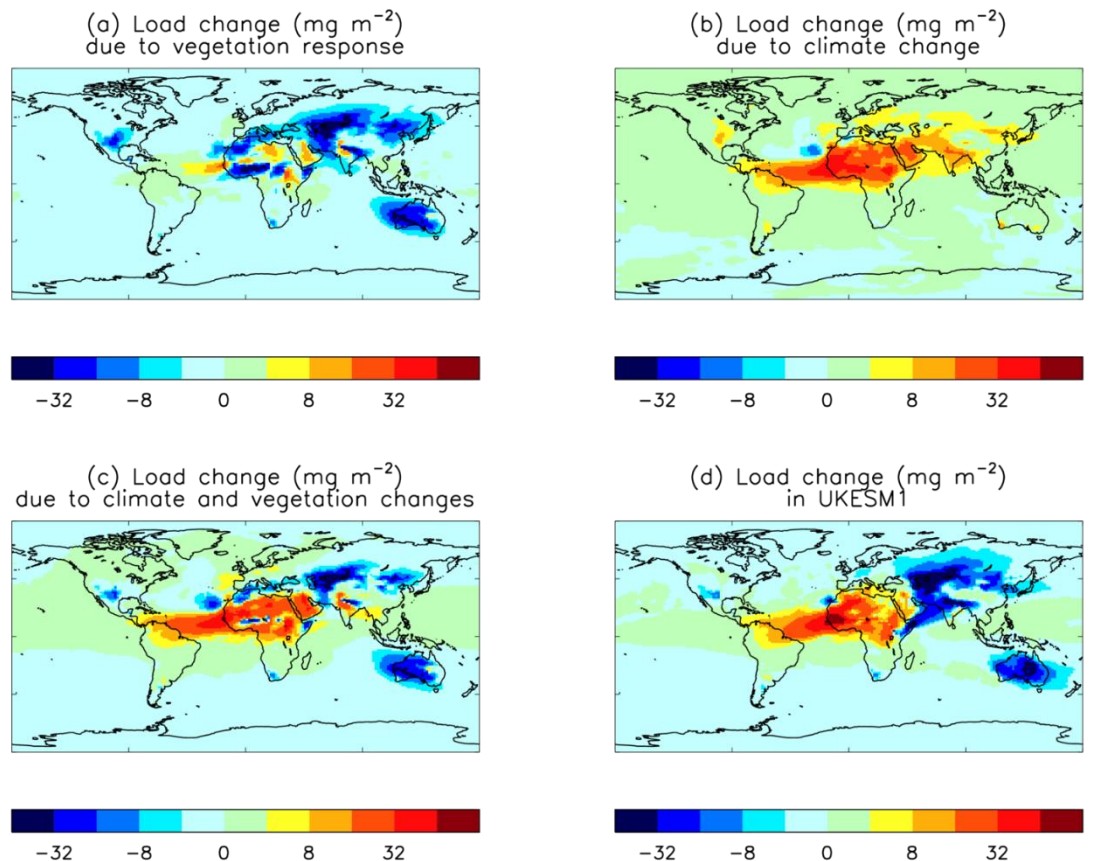

**Figure 9: Difference in dust load (mg m-2) from pre-industrial to the present day due to changes in vegetation (from A_UK_PI_V2014 – A_UK_PI) , due to changes in climate and land-use (from H3_PD – H3_PI), the sum of these two (A_UK_PI_ V2014 – A_UK_PI)+ (H3_PD – H3_PI), and in the full earth system model (from UK_PD – UK_PI).**

Results are summarised in Table 3 and Figures 11 and 12. UKESM1 simulates a reduction of 23% in total dust load from the present day to the end of the SSP5 scenario, with the greatest reductions from the major deserts, and only small areas experiencing increases (Fig. 11). In contrast the global load in HadGEM3-GC3.1 decreases by only 4%, with smaller local changes than in UKESM1 and reduced emissions from some deserts almost completely balanced by increases elsewhere (Fig. 12). The equivalent results for SSP2-4.5 are reductions of 19% for UKESM1 and 7% for HadGEM3-GC3.1. In each model the pattern of load changes from the present to the end of SSP2-4.5 have very similar geographical distribution to the

equivalent SSP5-8.5 results. This is also true of the main drivers of dust emissions: soil moisture, source areas and windspeed. The slightly larger decrease in dust load in H3_S2 compared with H3_S5 is due to minor differences in the residuals from the cancellation of areas of positive and negative change. Given the similarities in the patterns of change of dust and its drivers in both pathways, the following analysis will focus only on the SSP5-8.5 experiments.

| Experiment | Source Area Fraction | Load (Tg) | ToA SW (W m$^{-2}$) | ToA LW (W m$^{-2}$) | ToA Net (W m$^{-2}$) | Surface SW (W m$^{-2}$) | Surface LW (W m$^{-2}$) | Surface Net (W m$^{-2}$) |
|---|---|---|---|---|---|---|---|---|
| H3_PD | 0.458 | 15.01 | -0.460 | +0.164 | -0.296 | -0.688 | +0.338 | -0.350 |
| H3_S2 | 0.486 | 13.95 | | | | | | |
| H3_S5 | 0.485 | 14.47 | -0.440 | +0.150 | -0.289 | -0.652 | +0.291 | -0.361 |
| UK_PD | 0.255 | 19.54 | -0.280 | +0.194 | -0.086 | -0.556 | +0.388 | -0.168 |
| UK_S2 | 0.237 | 15.74 | | | | | | |
| UK_S5 | 0.221 | 15.07 | -0.191 | +0.143 | -0.048 | -0.400 | +0.269 | -0.132 |

**Table 3: Dust source area fraction, atmospheric load and DRE from simulations of present-day and future climates. (NB DRE diagnostics were not available for the H3_S2 and UK_S2 experiments.)**

Comparison of the changes in the drivers of dust emissions helps reveal the causes of the differences between the dust changes in the two models. Source area changes and associated vegetation changes are shown in Figure 10. In UKESM1 the dynamic vegetation responds to the warming climate and enhanced CO2 by producing increased growth. Areas of bare soil are colonised by grasses, whilst existing grassland is taken over by shrubs and trees. Land-use changes produce the opposite effect: areas of trees are lost to crops (grasses), most notably in sub-Saharan Africa, though this effect is small compared to the climate-driven vegetation changes. The net result is a decrease in bare soil, particularly in mid-latitudes. HadGEM3-GC3.1 vegetation only includes the land-use changes, which are represented as changes to the Tree, Shrub and Grass plant functional types. The bare soil fraction is constant, though source areas do show systematic change as dust is produced from seasonally vegetated grass and shrub areas in that model.

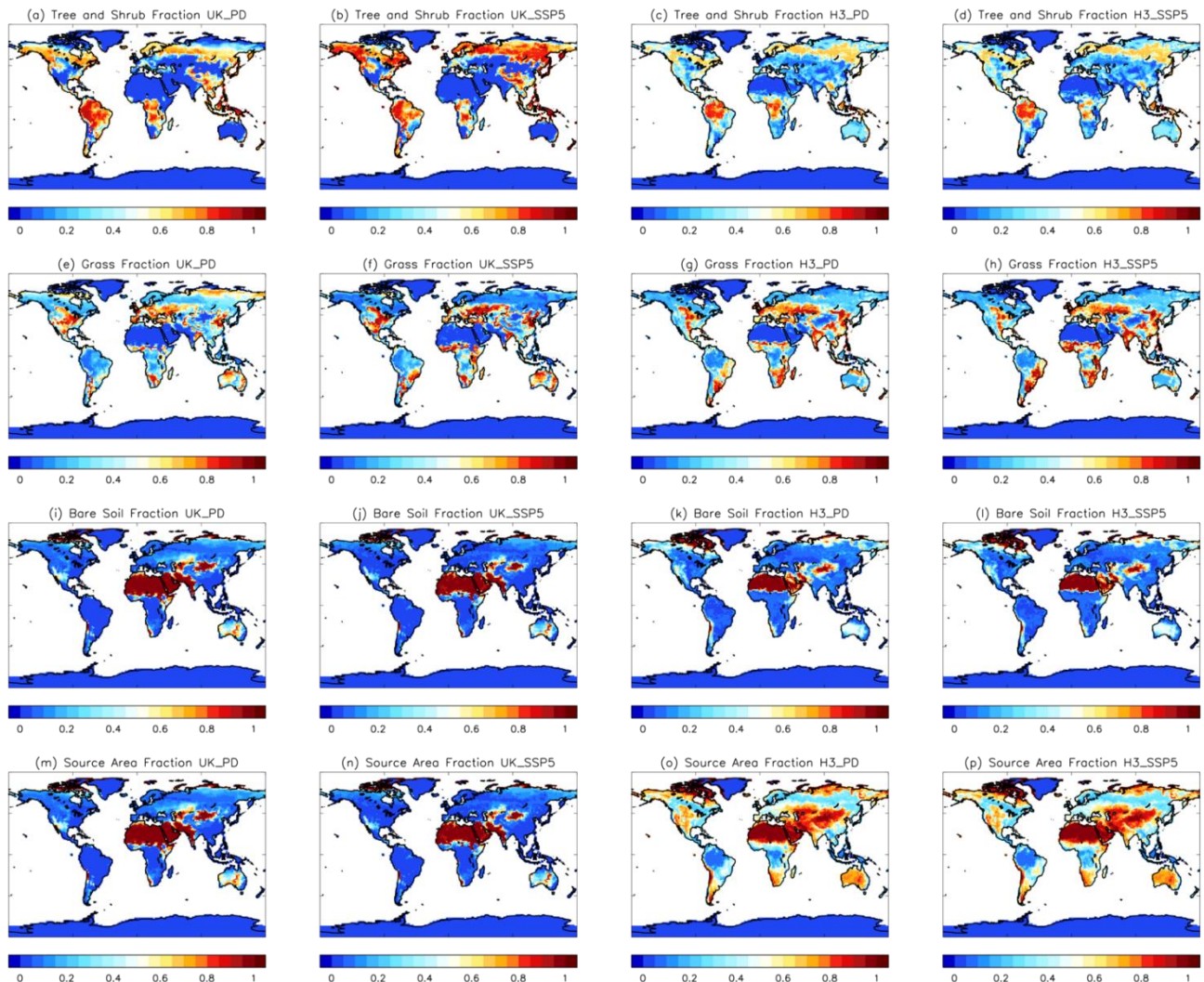

**Figure 10: Grass fraction, tree and shrub fraction, bare soil fraction and dust source area fraction from UK_PD, UK_SSP5, H3_PD and H3_SSP5. (The H3_PD and H3_SSP5 source areas include seasonal sources.)**


Figures 11 and 12 show the changes in dust fields and the drivers of emissions, simulated by UKESM1 and HadGEM3-GC3.1 respectively. The top-level soil moisture in UKESM1 is seen to be reduced in all areas except deserts, giving a global mean loss of 12%. A similar reduction of 15% occurs in HadGEM3-GC3.1 suggesting that this effect is mostly a response to the changing climate, mediated by evaporation and somewhat mitigated by the reduction in bare soil fraction in the earth

system model.

In UKESM1 the lowest-level windspeed over land is reduced in northern mid- and high-latitudes as well as in Australia and southern South America, but is increased in parts of the tropics, particularly tropical South America, though not in desert

regions. HadGEM3-GC3.1 exhibits similar increases in the tropics, but the mid- and high-latitude reductions are much smaller. This indicates that the tropical increases in windspeed are likely to be primarily a response to the changing climate,

whilst the reductions likely are due to increased roughness produced by vegetation growth in the corresponding areas in UKESM1.

The increase in dust emissions at high latitudes in HadGEM3 is driven predominantly by the reduction in soil moisture and at low latitudes by the increases in windspeed, whilst the reduced emissions from arid areas are caused by the slightly reduced windspeeds and moister soil in those regions. These processes also occur in UKESM1, but the vegetation driven

loss of bare soil and larger reduction in windspeed have a greater impact, resulting in enhanced reduction of emissions, with emissions increases occurring in only a few small regions. Whilst there is a global near balance of dust load increases and decreases in HadGEM3-GC3, UKESM1 simulates a global load reduction of 23% by the end of SSP5-8.5.

The difference in dust responses between the models is much larger than the difference in dust responses between the pathways. HadGEM3-GC3.1 results suggest that the global dust burden dust will remain largely unchanged whatever socio-

economic pathway is followed, with increases mostly in North and South America and Australia balanced by decreases mainly in North Africa, Asia and Europe. In contrast, in UKESM1 the addition of the extra earth-system process, and particularly the interactive vegetation, results in projections of reduced global total load, with reductions from most of the main desert regions and only a few small areas of increase in the tropics. The global load reduction of 23% associated with "Fossil-fuelled development" is somewhat larger than the 19% reduction of the "Middle of the road" pathway. We note the

non-linearity of this response to forcing, despite the linear response of the source area, and speculate this may be due to the dust emission process involving non-linear dependence on various factors which themselves may respond non-linearly to radiative forcing, and that dust feedbacks, which may enhance or limit emissions (eg Miller et al., 2004; Woodage and Woodward, 2014), could also introduce non-linearity. Our simulations suggest that the impact of the vegetation response on dust is larger than the direct impact of future climate change; and the differences due to including earth system processes in

the simulations is larger than the differences between pathways.

The reduction in load in UKESM1 results in a decrease in ToA DRE from UK_PD to UK_SP5 of 44%, though the absolute values are small (-0.086 and -0.048 W m$^{-2}$). The H3_S5 global mean TOA DRE of -0.29 W m$^{-2}$ is only 2% greater than the H3_PD value. In UKESM1 in particular, local changes are much larger than the global mean (Fig. 7).

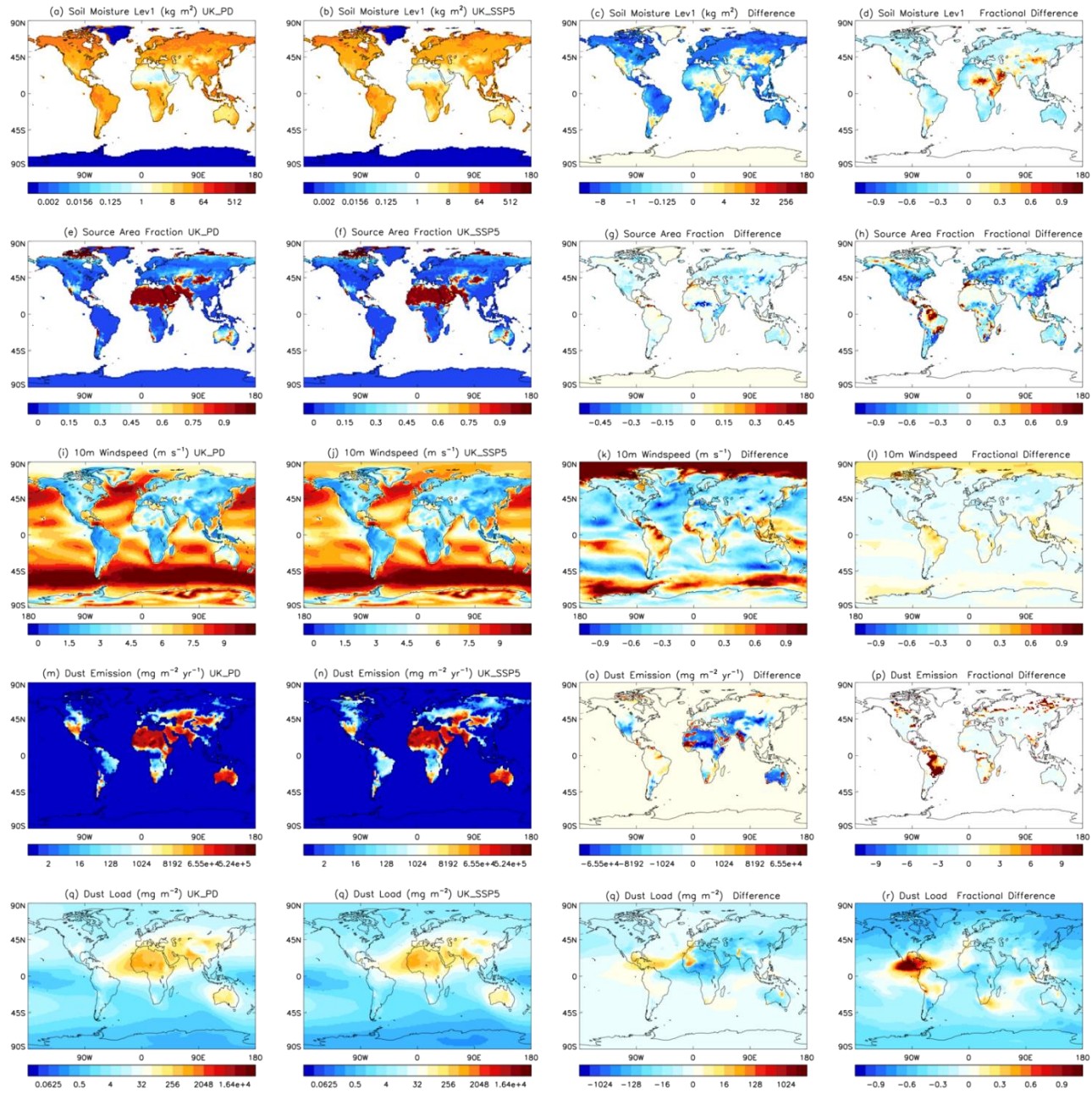


**Figure 11: Soil moisture in top layer, dust source area fraction, 10m windspeed, dust emission diagnostic and dust load from UK_PD and UK_SSP5, together with differences and fractional differences between experiments.**

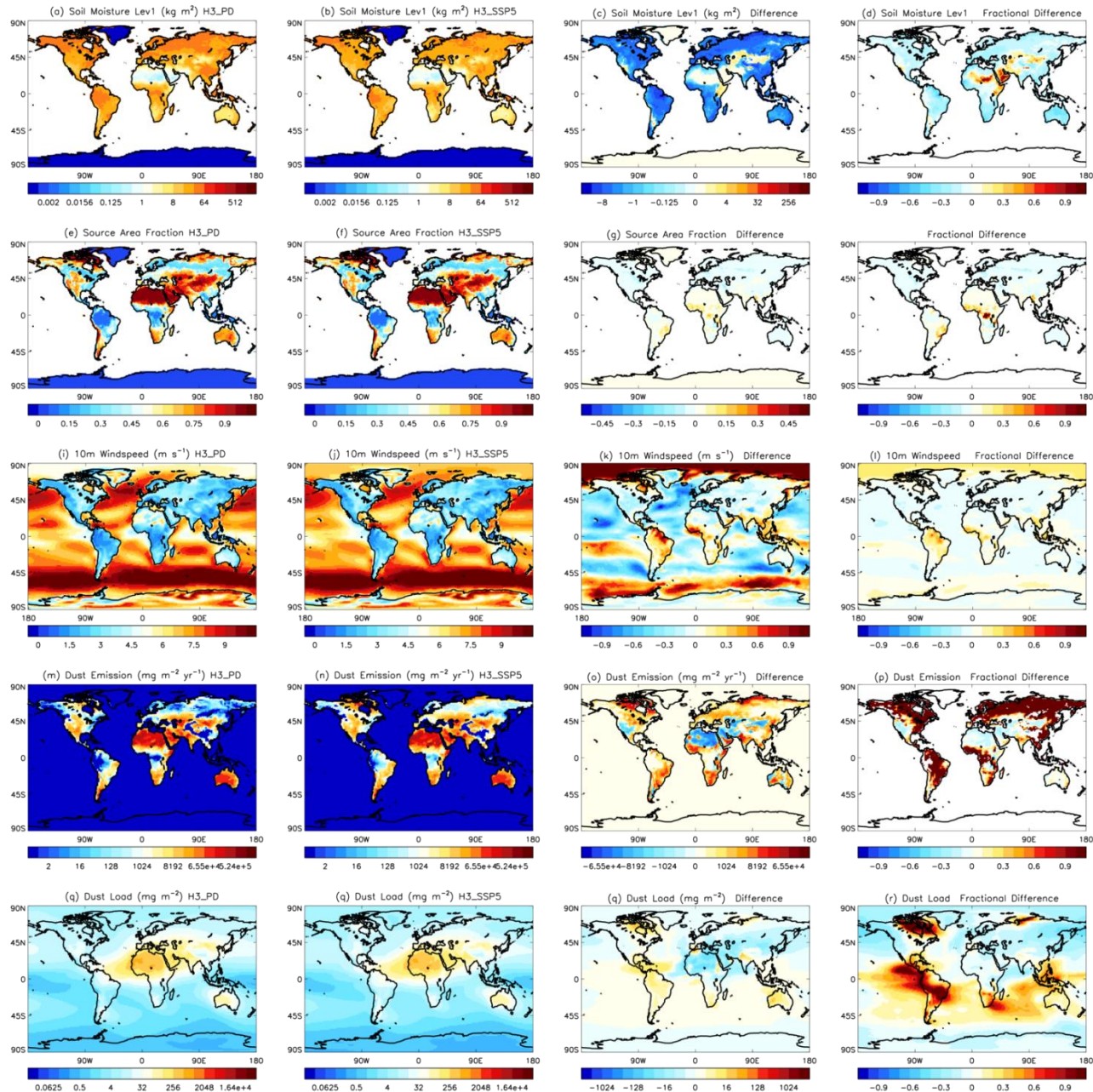


**Figure 12: as Fig. 11 but for H3_PD and H3_SSP5.**

# 7 Discussion


 The simulation of present-day dust in UKESM1 shows good agreement with most of the observations, though AoD is somewhat low.  The performance is generally comparable with that of HadGEM3_GC3.1, despite the extra complexity of the additional processes and feedbacks in the earth system model.  The tuning of the lai_min term for grasses in JULES, which limits the rate at which grasses can spread, was an important factor in this, as it improved the simulation of bare soil

fraction in source regions.  Some degradation in model performance is to be expected as a cost of the increased complexity, which is included to allow investigation of the feedback processes (Jones et al., 2011).  Based on the evaluations performed, it was not found necessary to use preferential sources to add further restrictions to the dust emissions in UKESM1 compared to HadGEM3-GC3.1, so the dust was able to respond to climate-driven changes without any potentially unrealistic constraints.  It is of course possible that other, as yet un-evaluated, properties of the present-day simulations, such as

frequency of occurrence of high dust optical depth, might be improved by the use of preferential sources.

Tuning of the dust scheme in the final UKESM1 configuration was limited by time constraints, as is often the case with the development of a model which includes dust.  Further work might possibly have resulted in an alternative choice of settings.  For example, in the UKESM1.1 model configuration dust was re-tuned to give better agreement with observations of AODs, though at the expense of agreement with other observations of size distribution and of concentrations in some areas

(Mulcahy et al., 2022).

The low bias in AOD in the Saharan plume is the main weakness of the UKESM1 tuning settings, though as AOD depends on dust optical properties as well as on size distribution and concentrations it is unclear quite how much of this bias is due to tuning.  Whilst optical properties are based on observations, they may well not be representative for all Saharan dust.  In particular, dust from the Bodele is made up predominantly of diatomite, which has different physical properties compared with dust from other Saharan sources (Todd et al., 2007).  The size distribution was shown to agree well with observations.

with dust from other Saharan sources (Todd et al., 2007).  The size distribution was shown to agree well with observations.

The simulation of the Saharan plume is also affected by model resolution.  Up to half the dust emissions from West Africa may be associated with cold pool outflows from moist convection (haboobs), particularly in summer when this fraction is most important (Caton-Harrison et al., 2019 and 2020), but the parametrised rather than explicit convection used in the N96 model renders the simulation of these events very difficult (Marsham et al., 2011; Knippertz and Todd, 2012), likely leading

to underestimation of emissions in that area.  Emissions from the Sahel are probably also underestimated for similar reasons. The omission of dust ageing through chemical processing has the effect of increasing lifetime and therefore increasing dust concentrations remotely from sources, and though this is not a very large effect, the inclusion of ageing could impact the choice of tuning settings, allowing higher Saharan AoDs without too much over-estimation of concentrations in other areas. A higher present-day AoD over the Sahara would likely be associated with larger DREs in both the SW and LW, assuming

the size distribution was unchanged.  These would still largely cancel, but the net DRE would probably be larger, as would the projected forcings for 2100.

The results demonstrate the crucial role of the particle size distribution in the simulation of the 3D dust distribution and particularly the radiative effects. Comparison of H3_PD with H3_TUK_INSS_NL (and H3_TUK_INSS) shows that quite small shifts in the size distribution towards coarser particles can change the balance of global mean SW and LW DREs so that they almost completely balance, giving net DREs close to zero The particle size range is the same in all cases, with a maximum diameter of 63μm, which is larger than many other climate model dust schemes (Huneeus et al, 2011; Checa-Garcia et al, 2012), so the DREs from HadGEM3-GC3.1 already include a relatively large LW effect, and hence relatively small net DREs.

The DRE results from UKESM1 are consistent with the estimates of Kok et al (2017), who find that most models underestimate the size of atmospheric dust compared with measurements, and use analysis of size and other observations to constrain model estimates of dust ToA DRE to a range between -0.48 W m-2 and +0.2 W m-2. This compares with a range of results from -0.56 W m-2 to +0.01 W m-2 reported in Forster et al. (2007), and more recent studies have also fallen within the latter range (Miller et al., 2006; Balkanski et al., 2007, Mahowald et al.,06; Albani et al., 2014), though Scanza et al. (2015) have a slightly more positive ToA DRE of +0.05 W m-2. Note that a small net ToA DRE does not necessarily imply a minimal effect on climate. Even globally homogeneous LW–SW compensation has been shown to affect climate (Tilmes et al., 2016), and in the case of dust regional effects will be important. One mechanism for this is the perturbation of the hydrological cycle by the dust DRE (e.g. Miller et al., 2004; Wilcox et al., 2010; Woodage and Woodward, 2014; Miller et al., 2014).

The response of dust to climate change in each of the models is very different, predominantly due to the direct and indirect impacts of the vegetation changes in UKESM1 which are not included in HadGEM3-GC3.1. This suggests that in the future the main anthropogenic impact on dust may be via the change in vegetation consequent on fossil-fuel driven emissions rather than through the changes in climate variables, whilst changes in land-use have a smaller effect. The differences between the models are greater than the differences between the SSP2 and SSP5 simulations from either model. These results illustrate the importance of including earth system processes when simulating the response of dust to climate change: indeed a realistic future dust simulation may not be possible without including vegetation changes and its effects.

Dust is particularly sensitive to the interactively simulated vegetation in an earth system model. Marginal areas for vegetation at the edges of deserts may be difficult for a vegetation scheme to simulate realistically but are potentially strong dust sources. Also, where vegetation is lost not only do source areas increase, but the consequent reduced roughness may lead to increased windspeed, and hence via increased evaporation to reduced soil moisture- changes which tend to enhance dust emissions. However, this effect is small in our climate change studies as vegetation tends to grow rather than die back in response to increased $CO_2$ and a warmer climate."

A feature of the pattern of emission changes, particularly in UKESM1, is the location of areas of large increase very close to areas of large decrease, with a sharp boundary between them. This demonstrates the sensitivity of the dust to the balance of the various driving fields, as well as the high sensitivity to windspeed. Even modest biases in driving model fields could produce shifts in such balances, and the areas of sharp gradients in dust changes are unlikely to be captured correctly by any

current schemes and present a challenge for future dust modeling. There are, inevitably, considerable uncertainties associated with such simulations as these. Biases in the driving model can have a significant impact on the dust simulation. Dust production is highly sensitive to friction velocity, but emission only occurs when this exceeds a threshold which depends on soil moisture, hence biases in model windspeed, precipitation or soil hydrology can have a large impact on dust.

Feedbacks, such as those between soil moisture, vegetation cover and windspeed can exacerbate the effects of biases. Precipitation biases have been noted in both models, particularly along the intertropical convergence zone (Williams et al., 2018; Seller et al., 2019), and these impact soil moisture. In UKESM1 the precipitation biases also affect vegetation, with lack of monsoon rain resulting in lack of C3 grass and excessive bare soil in India and the Sahel (Sellar et al., 2019). The model resolution limits the realistic representation of some processes, such as mesoscale convective systems which results in

biases in precipitation and dust mobilisation in the Sahel. These biases could be exacerbated by dust - precipitation feedbacks, as demonstrated by Yoshioka et al (2007). Biases in the bare soil fraction also impact emissions, an effect which will be larger in UKESM1 due to the use of the interactive vegetation scheme. Transport and deposition are also affected by driving model biases, particularly in the circulation and precipitation.

Simplifications, uncertainties and missing processes in the dust scheme will all produce biases and contribute to uncertainty

in the results. Seasonal sources are excluded from the scheme because they depend on the distributions of various plant types, and it was not clear whether these could be simulated sufficiently accurately in the ESM. Comparison of experiments H3_TUK_INSS and H3_TUK_EXSS indicates that without other changes the omission of seasonal sources would have reduce load by approximately 10%, though we have not assessed the realism of dust production from such sources in HadGEM3-GC3.1. The emission scheme ignores some factors, including re-entrainment and the effects of surface crusting

and surface geomorphology. The global tuning terms can only compensate for the effects of temporal and spatial averaging very approximately and the impact of heterogeneous, short term and sub-grid scale phenomena such as gustiness cannot be well represented. There are many uncertainties associated with deposition, as the role of such factors as electrical charging, electrophoresis, diffusiophoresis and rear capture in below-cloud scavenging is not yet fully understood. The treatment of deposition here is relatively simple, and in-cloud scavenging is ignored as it is a relatively small effect compared with

below-cloud scavenging for insoluble dust. Dust ageing through chemical processing is also ignored. The inclusion of ageing and in-cloud scavenging would be likely to result in lower concentrations remotely from sources. The use of a uniform set of refractive index data for all dust will result in biases in the DRE (Scanza et al., 2015). Lack of observational input data, such as global surface geomorphology, limits the representation of dust. There is also a paucity of data for validation of dust on climate timescales.

The accuracy of the estimates of the response of dust to climate change is difficult to assess. Any biases, uncertainties and feedbacks in the present-day simulations could well change with changing climate, and will inevitably impact the estimates of the dust responses and feedbacks. Factors associated with model resolution, driving fields, limitations of the dust scheme, missing processes or missing feedbacks could all be important. There is some evidence that a climate model may be unable to simulate the observed variability in dust over decadal timescales (Yoshioka et al., 2007; Mahowald et al., 2010), which

would be indicative of missing feedbacks and could result in the underestimation of changes in future projections. We have not analysed the variability of the modelled dust compared with observations, though UKESM1 does exhibit higher dust variability than HadGEM3-GC3.1 probably due the additional processes and feedbacks in the ESM.

Despite the inherent limitations and uncertainties, these results can provide much useful information, providing that the caveats are borne in mind. Further work is required to provide the missing observations, to study the processes affecting
dust, and to improve dust schemes and the models driving them, in order to enhance understanding and improve the simulation of the effects of climate change on dust.

## 8 Conclusions

The dust scheme used in UKESM1 and HadGEM3-GC3.1 is a development of the Woodward (2001) scheme with an improved emissions parametrisation. It was initially used in HadGEM3 and then re-tuned for UKESM1. Seasonal sources
were also de-activated in UKESM1, reducing dependence on multiple plant types in the interactive vegetation scheme whilst having only a small effect on load (less than 10% in HadGEM3-GC3.1). Evaluation of the UKESM1 present-day dust simulation showed good agreement with observations, comparable with that of HadGEM3-GC3.1. This is particularly encouraging given the additional uncertainties produced by the extra processes and feedbacks within the earth system model, and the fact that no extra constraints, such as preferential sources, were applied to limit the dust emissions in UKESM1.
The differences between global mean UKESM1 and HadGEM3-GC3.1 present-day DREs have been shown to depend on the change in size distribution consequent on re-tuning as much as on the change in driving model. The change in load due to re-tuning had a lesser impact, and the activation of seasonal sources produced only a small effect. This demonstrates the importance of the simulation of size distribution, which impacts the global dust concentrations through the size dependent deposition processes and additionally impacts DREs through the size dependent radiative properties. The magnitude of this
effect will depend on the choice of refractive index data.

The response of the dust under future socio-economic pathways is highly model dependent. In HadGEM3-GC3.1 the climate response produces drying of moist soils which tends to increase emissions slightly, but this is balanced by a small reduction in windspeeds in major source regions leading to the total emissions and load remaining almost unchanged between the present day and 2100. Whilst these processes also occur in UKESM1, a greater impact on dust comes from the
vegetation response, which is simulated interactively in this model. Enhanced vegetation growth produces a decrease in bare soil leading to further reductions in windspeed, both of which result in lower emissions and load. The differences between the models are greater than the differences between the SSP2-4.5 and SSP5-8.5 pathways and, though there are some considerable uncertainties associated with these results, they indicate the importance of including the vegetation response in projections of dust in future climates.
These results provide useful new information about the interactions between dust and climate. They highlight the need to represent the full dust size distribution as realistically as possible, and indicate that the effect of doing this may be to reduce

the global mean net dust DREs at ToA to a value close to zero (though impacts on climate may be larger, particularly on a regional scale). They also show the importance of including earth system interactions in dust simulations over climate timescales, as the greatest driver of dust change may be vegetation changes and the consequent modifications to source

areas, soil moisture and windspeed.

**Code Availability.**

Information on the UKESM1 configuration and its components and the pre-requisites for using it may be found at http://cms.ncas.ac.uk/wiki/UM/Configurations/UKESM (last access 12 Mar 2022). Due to intellectual property rights we

cannot provide the source code or documentation papers for the UM or JULES. A number of national meteorological services and research organisations use the UM in collaboration with the Met Office to undertake research, produce forecasts, develop the UM code and build and evaluate Earth system models. The UM is available for use in this way under licence; see http://www.metoffice.gov.uk/research/modeling-systems/unified-model (last access 12 March 2022). JULES is available under licence free of charge for research purposes; see https://jules-lsm.github.io/access_req/JULES_access.html

(last access 12 Mar 2022).

**Data Availability.**

The simulations used in this work are based on CMIP6 simulations which are archived on the Earth System Grid Federation (ESGF) node https://esgf-node.llnl.gov/projects/cmip6/ (last access: 16 March 2022). The model source ID is UKESM1-0-LL for UKESM1 and HadGEM3-GC31-LL for HadGEM3-GC3.1. AERONET AOD data are available from

https://aeronet.gsfc.nasa.gov/ (last access: 16 Mar 2022)

**Author Contributions.**

SW was primarily responsible for the development of the dust code and its application in UKESM1, and ran the majority of the experiments and performed the analysis. AS was scientific manager for UKESM1, coordinated the implementation of the dust code in the ESM and provided invaluable suggestions and comments throughout. YT implemented the dust settings

into the UKESM1 model. MS wrote the code for dust deposition into the ocean. AY provided data and advice on deposition of dust to the ocean. ER ran the UK AMIP experiments and advised on their use. AW advised on the lai_min vegetation changes in UKESM1.

**Competing Interests.**

The authors declare that they have no competing interests.

**Acknowledgements.**

This study would not have been possible without the work of many staff in the UKESM1 Core Group and the Met Office who developed the UKESM1 model and set up and ran the CMIP6 experiments. We are also grateful to all the researchers who generated the observational data we have used, including those of the AERONET community and the University of Miami Aerosol Network. We thank Claire Ryder for providing information about the FENNEC campaign.

**Financial Support.**

This research was supported by the Met Office Hadley Centre Climate Programme funded by BEIS and Defra (SW,AS,YT,ER,AW) as well as NERC Grant NE/N017978/1 (MS) and the EU Horizon 2020 CRESCENDO project, Grant 641816 (AY).

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
