# Peer review of "The simulation of mineral dust in the United Kingdom Earth System Model UKESM1."

_Atmospheric Chemistry and Physics, 2022_

## Author Comment (AC1)

We would like to thank Ron Miller and the anonymous reviewer for their very helpful comments and suggestions, as well as some interesting new references. The paper has been updated in the light of these.

In the section below, reviewers' comments are in blue, our responses in black and text changes in green. Page and line numbers refer to the revised text, unless otherwise stated.

Response to comments by Ron Miller:

Major Comments:

1. Differences between the two model versions with (UKESM1) and without (HadGEM3-GC3.1) the Earth System feedbacks are partly the result of differences in tuning of dust model parameters, rather than differences in the range of physical processes represented by each model. These parameters are identified in the article (line 132) but little information is given about the observations used for tuning. More discussion of the tuning is important because differences in the direct radiation effect (DRE) between UKESM1 and HadGEM3-GC3.1 are attributed to differences in tuning and the impact upon the present-day size distribution (line 563).

Section 3.3 "Dust Tuning" has been re-written to include further information about the observations and the tuning process for UKESM1. As the text now explains, we did not re-tune HadGEM3-GC3.1, but took the standard version of the model which allowed us to use results from the CMIP6 simulations. We now mention that the observational datasets were the same as those employed for the model evaluation and described in Section 5, and the sites from the University of Miami network are now listed in the caption of Fig. 2.. The new section 3.3 reads:

**"3.3 Dust Tuning**

[revised manuscript text omitted]

2. The agreement of both model versions with observations is described as "encouraging" (line 560) and used to question the necessity of including preferential sources that are absent in UKESM1 but commonly present in other models (cf. also line 490). However, the importance of preferential sources is typically demonstrated using the frequency of occurrence (FoO) of AOD above a certain value, and noting that regions of frequent dustiness (high FoO) are highly localized, suggesting that dust emission itself is confined to limited regions (Prospero et al. Rev. Geophys. 2002). This is the motivation for the preferential source maps of Ginoux (2001, 2012) along with Zender (2003) and Tegen (2002), even if their physical criteria that identify sources vary. For the authors to demonstrate that their model does not need preferential sources, they would have to show that their FoO, which is not diagnosed for either model, is consistent with the observed values. The authors need not diagnose FoO, but they should note that their choice of observations for evaluation does not distinguish the effect of their preferred source omission.

We acknowledge that an evaluation of FoO to show regions of frequent dustiness could reveal shortcomings in the UKESM1 simulation, that might well be ameliorated by the use of a preferential source term. For various reasons we decided not to employ preferential sources in UKESM1 unless the dust simulation was significantly worse than in HadGEM3-GC3.1 and additional restriction on source areas was required to achieve a similar level of performance. The dust in the standard HadGEM3-GC3.1 model (which we used "as is", as now mentioned in Section 3.3) does not include preferential sources, and adding them to the UKESM1 dust would have rendered comparisons between the two models, and hence the investigation of the effects of the earth system processes, more difficult. Introducing new elements to the scheme in the HadGEM3-CG3.1 configuration was outside the scope of the project. In addition, as the mechanism for preferential sources is not fully understood (with hydrological, topographical and geomorphological processes all suggested), there is a danger that a preferential source term chosen to match present-day observations might be "right for the wrong reasons", and could add uncertainty or biases to projections of future climate, particularly with different vegetation cover. An assessment of FoO would have been difficult at the time of development, as at that point there was no mechanism for driving or nudging UKESM1 with reanalyses or other observations, which would ideally be required for this type of analysis.

The comments suggesting it was encouraging that preferential sources were not needed were intended to indicate that despite the extra complexity of the ESM, and particularly the use of interactively modelled vegetation, the dust simulation, as we evaluated it, was not significantly worse than that in the more strongly prescribed HadGEM3-GC3.1 model. The text has now been changed to

(P28, L 582, former line 490):

"Based on the evaluations performed, it was not found necessary to use preferential sources to add further restrictions to the dust emissions in UKESM1 compared to HadGEM3-GC3.1, so the dust was able to respond to climate-driven changes without any potentially unrealistic constraints. It is of course possible that other, as yet un-evaluated, properties of the present-day simulations, such as frequency of occurrence of high dust optical depth, might be improved by the use of preferential sources."

(P31, L 688, former line 560):

"This is particularly encouraging given the additional uncertainties produced by the extra processes and feedbacks within the earth system model and the fact that no extra constraints, such as preferential sources, were applied to limit the dust emissions in UKESM1."

3. The future experiments suggest that the expansion of vegetation in the late 21C will reduce dust emission, but it is essential to show that this feedback is simulated with the correct amplitude. Some observations suggest that dust models are failing to simulate large decadal variations in source strength and dust emission (e.g. Yoshioka et al J. Climate 2007, Mahowald ACP 2010). These studies should be noted, because they raise the possibility that the change in dust between the pre-industrial and present day is larger than simulated by current models, suggesting that the models and their projections of future dust emission are lacking an important dust-climate feedback.

We agree that it is very important to establish the magnitude of changes in dust emissions in a future climate. Our simulations can only provide one step towards that goal and much further work is needed. A detailed analysis of our present-day results, similar to those performed by Yoshioka et al. or Mahowald et al., is beyond the scope of this work, but a paragraph discussing the issue in the context of those studies has been added to the Discussion (P30, L670):"

"The accuracy of the estimates of the response of dust to climate change is difficult to assess. Any biases, uncertainties and feedbacks in the present-day simulations could well change with changing climate, and will inevitably impact the estimates of the dust responses and feedbacks. Factors associated with model resolution, driving fields, limitations of the dust scheme, missing processes or missing feedbacks could all be important. There is some evidence that a climate model may be unable to simulate the observed variability in dust over decadal timescales (Yoshioka et al., 2007; Mahowald et al., 2010), which would be indicative of missing feedbacks and could result in the underestimation of changes in future projections. We have not analysed the variability of the modelled dust compared with observations, though UKESM1 does exhibit higher dust variability than HadGEM3-GC3.1 probably due the additional processes and feedbacks in the ESM."

Minor Comments:

22 "0.289 W m-2" missing minus sign?

This has been corrected (P1, L22).

35 "Climate models have long included dust schemes (e.g. Tegen and Fung, 1994...)" I'm not sure how 'climate model' is defined but Tegen and Fung describes an offline chemical transport model driven by GISS AGCM winds. The integration of a dust scheme into the actual GISS AGCM was described by Tegen and Miller JGR 1998.

The citation has been corrected. "Tegen and Fung, 1994" replaced by "Tegen and Miller, 1998", (P2, L35).

55 "such as the size distribution" I find this confusing because it is a consequence of model differences rather than a model difference per se. I suggest limiting the description to the latter. I am guessing that the HadGEM3- GC3.1 experiment holds vegetation fixed? Are there other differences that might be important? (Some of the discussion below that addresses my comment could be moved here.)

The sentence has now been re-worded in an attempt to clarify this (P2, L53):

"We then compare the UKESM1 simulations with parallel results from HadGEM3-GC3.1 and investigate the causes of the differences: in particular the consequences of including interactive vegetation and other earth system processes and the effects of the changes to the dust scheme settings, including the impacts of re-tuning on load and size distribution and of excluding emissions from seasonal sources."

90 "inhibited" Do you mean 'prevented'?

Inhibit and prevent are synonymous in British English ( eg  https://www.collinsdictionary.com/dictionary/english/inhibit), but we have changed the word to avoid confusion for the American readership (P3 L91)

93 "equation (1) was derived from instantaneous measurements at single locations." Is this true? I believe that equation (1) is based upon averaging over a time scale related to the passage of several turbulent eddies, corresponding to at least a few minutes.

This has now been changed to (P4 L 94).:

"derived from shorter timescale measurements "

94 "Corrections are therefore needed..." The parameter k1 is uniform with space and time, even though rapid fluctuations might be specific to certain conditions. e.g. Lunt JGR 2002 and Cakmur JGR 2004 relate gustiness to the surface sensible heat flux, which is typically strongest at midday and within arid regions. This use of k1 should be discussed in this context.

A paragraph on this topic has been included after equation (2) (P4 , L98):

"Ideally such a correction would be spatially and temporally variable to account for rapid fluctuations which might be specific to certain conditions; for example Lunt and Valdes (2002) and Cakmur et al. (2004) related gustiness to surface sensible heat flux, which is typically strongest at midday and within arid regions. These studies suggest that introducing a representation of gustiness can have a large impact and can improve dust simulations, though the magnitude and spatial distribution of the effect appears to be strongly dependent on the parametrization used. Cakmur et al (2004) showed that a parametrization using a linear combination of the gustiness due to dry and moist convection and turbulent kinetic energy produced considerable improvements in modelled optical depth in many areas. The use of a global tuning term rather than a realistic representation of gustiness will introduce biases to our simulations, with friction velocity over-estimated in some areas at some times, and under-estimated in others. Though the higher resolution of our models compared to those of Lunt and Valdes (2002) and Cakmur et al. (2004) (1.875°x1.25° compared with 3.75°x2.5° and 4°x5° respectively), together with the shorter timestep (20 mins compared with 30 minutes interpolated from 6-hourly input and 1 hour respectively) should allow a somewhat better representation of the smaller scale and more variable processes, many phenomena important for dust generation are still sub-grid scale."

98 "Dry threshold friction velocity (U*td) values were obtained from Bagnold (1941)" Could you give more information about your threshold dependence upon diameter? My memory is that Bagnold proposed a threshold that increased with particle diameter, whereas subsequent measurements showed a minimum near 75 um diameter and a subsequent increase of the threshold for smaller particles. (e.g. Greeley and Iversen 1985).

The Bagnold $U^*_{td}$ also reaches a minimum near 75 um.. The values used for each bin are now listed in the text (P4, L111):

"The values for each of the 9 horizontal flux bins are 0.85, 0.72, 0.59, 0.46, 0.33, 0.16, 0.14, 0.18 and 0.28 m s$^{-1}$."

128 "In UKESM1 the total marine dust deposition flux is passed into the ocean" During atmospheric transport, is there any processing of the iron in dust into bioavailable forms? (e.g. Myriokefalitakis GMD 2022)

Further explanation has been added (P5 L144):

"No chemical processing of dust in the atmosphere is represented: this is something we hope to include in future versions of the scheme. In UKESM1 the total marine dust deposition flux is passed into the ocean, where it is multiplied by a tuned constant to derive "bioavailable iron" (Yool et al., 2021) which is used by the MEDUSA ocean biogeochemical scheme (Yool et al., 2013) as a nutrient source for plankton growth."

161-163: please give more specific information about what observational criteria were used that resulted in different values of the tuning parameters between the two models.

HadGEM3-GC3.1 is a widely used, standard model configuration. It also forms a central component of the UKESM1 model, but the sensitivity of the dust scheme to the driving model fields means that re-tuning had to be carried out when the UKESM1 model was developed. However, re-tuning the scheme within HadGEM3-GC3.1 was outside the scope of the project. This is now explained in the re-written version of section 3.3 "Dust Tuning", together with information on the observations and approach to tuning used for UKESM1 (see text after response to major comment (1) above).

194 "the global load in UK_PD is 30% higher than in H3_PD." Could you comment on regional differences (Fig. 1c and d)? In key dust emitting regions like parts of the Sahel, the difference between model versions can be up to a factor of 2.

This paragraph has been rewritten in response to this and other comments. The final sentence addresses regional changes. It now reads (P8, L 235):

"Figure 1 shows dust load from the UKESM1 and HadGEM3-GC3.1 present-day simulations (UK_PD and H3_PD), together with the differences and fractional differences. Both models capture the expected global dust global dust distributions qualitatively well. The UK_PD global load of 19.5 Tg is 30% higher than the 15.0 Tg H3_PD value. These values span the AeroCom phase III mean of 16.6 Tg, and are within the range of 5.7 Tg to 22.3 Tg shown by participating models (Gliss et al., 2021). They are also consistent with the more observationally constrained estimate of Kok et al. (2017) for PM20 dust load of 23 Tg with a range from 14 Tg to 33 Tg. The greatest differences between the models are over the Sahel, India, the Middle East, Asian midlatitudes and Australia, and are predominantly due to differences in the bare soil and associated changes (see Section 5.6 and Fig. 8)."

195 "The UK_PD load of 19.5 Tg is in good agreement with the AeroCom mean of 19.2Tg (Textor et al, 2006)." A more observationally based estimate of the load is given by Kok et al. Nat. Geosci. 2017: 23 (14–33) Tg. This is strictly an estimate for PM20, but the addition of larger particles probably increases this value only slightly.

This has been rewritten (see previous response)

200 "dust-dominated AERONET" Note that AERONET measures AOD. Also, how were dust dominated sites identified? (Maybe move discussion near line 245 to here?)

This has been reworded (P9 L246):

"AERONET sites (Holben et al., 2001) chosen for data availability and dustiness based on the Angstrom exponent (see Section 5.3)."

207 "and the position of the Subtropical Front" Is this the atmospheric Subtropical Front? How is this identified? By the position of the subtropical jet?

Atmospheric subtropical front had been intended. The clause has now been removed (L 255) as it adds little to the sentence, and explanation of what definition was used would be cumbersome here.

209 "The similarity in the level of performance between UKESM1 and HadGEM3-GC3.1 is noteworthy" Could you quantify this performance, e.g. with a correlation coefficient for both surface concentration and AOD between the two models?

A sentence has been added earlier in the paragraph (P9 L 250):

"The correlation coefficients for the concentration and AOD data shown in Fig.2 are 0.89 for UKESM1 and 0.87 for HadGEM3-GC3.1."

225 "UKESM1 dust was tuned to give better agreement with the size distribution data from the FENNEC campaign (Ryder et al., 2013) than was shown in HadGEM3-GC3.1" Why wasn't the FENNEC data used to tune HadGEM3-GC3.1?

As now explained in Section 3.3, we took the standard HadGEM3-GC3.1 configuration as had been used for the CMIP6 simulations, and did not retune it. Our focus here was on UKESM1, and we used HadGEM3-GC3.1 to help understand the behaviour of the dust in the ESM. We had not been involved in the set-up of the HadGEM3_GC3.1 configuration, though we did design the dust scheme and originally implemented it in an earlier version of HadGEM and now in UKESM1.

See Text of Section 3.3 in response to major comment (1) above.

240 why is the UK_PD AOD consistently smaller than that due to H3_PD?

The lower UK_PD AOD at these sites is due to the difference in size distribution, as evidenced by a lower Angstrom exponent in the UK_PD simulation. We have added a paragraph to section 5.3 in which we compare the AODs and Angstrom exponents of UK_PD and H3_PD with the AERONET data (P11 L301):

"The observed multi-annual mean AOD at 440 nm meaned over these AERONET sites is 0.45, whilst UK_PD simulates 0.28 and H3_PD simulates 0.37. Comparison of the simulations shows that the lower AOD in UK_PD is due to a combination of lower dust optical depth (DOD), (0.13 in UK_PD and 0.19 in H3_PD) with lower optical depth due to other species (0.15 in UK_PD and 0.18 in H3_PD). The lower DOD in UK_PD is caused by the larger particle sizes in that model, as evidenced by the Angstrom exponent for the DOD (440-870) which is -0.06 in UK_PD and 0.23 in H3_PD, whilst the mean atmospheric load at these sites is higher in UK_PD: 540 mg m$^{-2}$, compared with 340 mg m$^{-2}$ in H3_PD. The Angstrom exponent for the total AOD (440-870) is 0.65 in UK_PD and 0.76 in H3_PD, compared with 0.33 in the AERONET observations. This indicates too low a coarse mode fraction in the simulations which suggests too little dust, as this is the dominant coarse mode species at these locations and the low simulated AODs show that there is no excess of fine mode aerosol. The slightly better agreement of UK_PD with the observed Angstrom exponent may be an indication that the particle size distribution is more realistic than in H3_PD, which would be consistent with the comparisons with FENNEC data, though with the caveat that the potential effects of different concentrations of other aerosol species at the AERONET sites cannot be ignored."

246 "Capo Verde off the west African coast" Does the AOD at this location include the effect of sea salt, which like dust, has large diameters and a small AE?

Sea salt is present at Capo Verde, particularly in the marine boundary layer, but dust dominates the coarse fraction (Ryder et al, 2018) which we know from the Angstrom exponent is the most important fraction for the AOD here. Fomba et al (2014) showed that mass fraction of PM10 from dust was was over twice that from sea salt sampled at the site over a five year mean. We have changed the sentence which mentions the AOD at Capo Verde( P12 L 328) to:

"Similar patterns are seen in the AOD at Capo Verde which is dominated by Saharan dust, though sea salt is also present at this site (Fomba et al., 2014, Ryder et al., 2018)."

259 "(MCS) produce similar levels of dust to those seen in the dry season." Please cite Caton-Harrison et al. JGR 2019 and 2020.

The citations have been added (P12 L319).

263 "N96 climate model" What is the approximate resolution of N96?

We have added (P12 L 324): "with a resolution of 1.85° x 1.25° "

277 "The high bias on the southwest side of the Himalayas and low bias on the northeast side suggest the model may be failing to transport aerosol over the steep orography there." Couldn't some of this bias be due to other aerosol species that contribute to the MODIS retrievals?

We agree, though the combination of the high bias on one side of the mountains combined with a low bias on the other side does seem particularly suggestive of a transport issue. We have added (P14 L346)

" or may be associated with other aerosol species."

297 "this collection of deposition" Albani et al. JAMES 2014 makes available a similar collection that acknowledges differences in the sampled particle size range. Is the size range from the Huneeus set used here uniform across all the sites?

Huneeus et al. acknowledge that they do not have size distribution data or measurement cut-off sizes for many of their sites, but argue that as their observations are at remote locations they are likely dominated by PM10. As we model particles up to 63μm diameter we might reasonably expect the modelled size range to span that of the observations at these remote sites. This situation is somewhat different from that of Albani et al, who included sites near sources which could have a significant coarse particle fraction, for validation of a model with a maximum particle size of 10μm. Nevertheless, it would be preferable to have particle size data available, as it is possible that low measurement cut-off points might have excluded some large particles, hence our caveat that "there are considerable uncertainties associated with the observations".

304 "noticeable bias is in Antarctica and the Southern Ocean" Is the overestimate of deposition related to the underestimate of surface concentration in these regions (Figure 2a)?

It seems reasonable that the over-estimated deposition in this region could have contributed to the under-estimated concentrations. However, the sites where deposition was over-estimated were on the opposite side of Antarctica to the stations where concentration was under-estimated, and the latter are in an area which is strongly influenced by the relatively local Patagonian dust, so we can't say for certain without further analysis.

306 "and hence roughness." How is roughness related to deposition?

This was a mistake: it should have read "may be due to overestimates in windspeed *or underestimates* in sea-ice cover". Deposition is calculated using a resistance analogue method (Woodward, 2001), in which deposition velocity is inversely dependent on aerodynamic resistance, which is itself inversely proportional to the drag coefficient for momentum and to the friction velocity, which both have roughness length dependence. We have now re-worded the sentence to clarify this (P17 L374) :

"This localised bias may be due to overestimated windspeed or to the underestimated sea ice cover which has been observed in UKESM1 (Sellar et al., 2019) and which would result in increased roughness and thus reduced aerodynamic resistance, leading to increased deposition velocity (Woodward, 2001)."

324 "being positive over the brightest surfaces" Is this an albedo effect as implied by the sentence? The positive forcing could also result from large LW emission toward the surface by the dust layer over bright surfaces where vegetation is sparse and emission is large.

The net surface DRE being "positive over the brightest surfaces" is just a simple description of the results as shown in Fig. 7(j), with no particular mechanism implied. It seems probable that both albedo and LW effects could be involved. Fig. 7(f) shows a large positive LW DRE over and downwind from the desert source areas; but it is only directly over the deserts that this dominates over the negative SW effect (Fig. 7(c)), as is particularly noticeable at the coast of N Africa in Fig. 7(j). However, we are here limiting ourselves to descriptions of the DREs, rather than exploring the mechanisms involved, which would make this section very much longer.

356 "feedback between dust DREs and emissions" see also Miller et al. (2004) doi: 10.1029/2004JD004912.

This reference has been added.

371 "SW DREs are approximately halved due to the size distribution change, mainly because of reductions in the load of finer particles in bins 2 and 3 (0.2–2.0 µm)..." It's hard to keep track which model is being referred to :) I'm guessing that the UK model has the larger particles? Given this fundamental difference in particle size, a figure describing it would be helpful.

An extra panel has been added to figure (3) to show the normalised global mean size distributions from UKESM1 and HadGEM3-GC3.1 simulations. This section has been reworded to try to improve clarity (P19 L449) :

"Change in size distribution consequent on re-tuning (Factor 1) is the factor responsible for the greatest absolute change in the net DREs, due to its large impact in the SW not being balanced by its much smaller LW effect. SW DREs are approximately halved in response to this size distribution change, mainly because of reduction in the number of finer particles in bin2 (0.2–0.63 µm) which have a strong SW effect. In contrast, LW DREs are only reduced by about 10%, chiefly because the alteration in the number of coarser particles is relatively small. In particular, the global number of bin 4 particles (2.0–6.3 µm), which dominate the LW effect, is almost unaltered, with increases in the Saharan plume and northern mid-latitudes balanced by reductions elsewhere, whilst the numbers of bin 5 and 6 particles (6.3–63 µm) which are somewhat less radiatively active are not greatly increased."

[Figure]

Figure 3: Dust volume size distributions: (a) normalised distribution from a fit to FENNEC mean, maximum and minimum data (Ryder,2013) and from a multi-annual June mean of UK_PD data meaned over a rectangular area 13W-4W, 21S-26N covering the FENNEC campaign region, for model levels 3 (approx. 96m) and 10 (approx. 770m), corresponding to the height range of most of the FENNEC measurements; (b) as (a) but for H3_PD; (c) global mean size distribution from UK_PD, H3_PD and H3_TUK_INSS_NL.

382 "6 The response of dust to changing climate" Maybe have two subsections: 6.1 for PI to PD and 6.2 for PD to the future?

We have followed this suggestion.

393 "The effect of climate change excluding the vegetation response (but including land-use change) is estimated from the difference between HadGEM3-GC3.1 simulations H3_PD and H3_PI." It seems potentially inconsistent to use the H3 model to diagnose the land-use effect in the UK model, given the different tunings. In fairness to the authors, the global averages in

Figure 9 seem to argue that the H3 model can be used to diagnose the effects of land use and climate change in the UK model, although regional differences between Fig. 9c and d can be significant (e.g. over India and parts of the Sahel).

The original reason for comparing the PI to PD changes UKESM1 with those in HadGEM3_GC3.1 plus A_UK_PI(+/-V2014NOLU) was to ascertain whether the dust responses to climate change and vegetation would dominate the signal, so that the two models could reasonably be compared, or whether the differences due to dust tuning or the impacts of other ESM feedbacks would be too large. As we wrote: "The similarities between the patterns of load change due to the combined vegetation and climate changes and due to the changes in UKESM1 is notable, giving confidence that dust changes simulated by the models are comparable even though the present-day dust is somewhat different." It therefore seemed reasonable to use HadGEM3-GC3.1 results in the exploration of the role of vegetation over the historical period. This approach meant that we could utilise existing results, instead of having to create a whole new configuration of UKESM1 without the interactive vegetation and then running long simulations.

466 "The global load reduction of 23% associated with "Fossil-fuelled development" is somewhat larger than the 19% reduction of the "Middle of the road" pathway." Why don't these changes scale linearly with the radiative forcing? (4.5 v. 8.5 W/m2)? This seems like an important question if the future change in dust is predictable.

The non-linearity of the response of dust to the radiative forcing is an interesting issue. Would a linear response necessarily be expected? We have added a sentence on this (P25 L554):

"We note the non-linearity of this response to forcing, despite the linear response of the source area, and speculate this may be due to the dust emission process involving non-linear dependence on various factors which themselves may respond non-linearly to radiative forcing, and that dust feedbacks, which may enhance or limit emissions (eg Miller et al., 2004; Woodage and Woodward, 2014), could also introduce non-linearity."

490 "It was not found necessary to limit dust emissions by imposing any sort of preferential source terms". See major comment 2.

Please see response to major comment (2).

510 "Even globally homogeneous LW–SW compensation has been shown to affect climate (Tilmes et al., 2016), and in the case of dust regional effects will be important." One reason is that the DRE at the surface, which is usually farther than the TOA value from zero, will perturb the hydrologic cycle (e.g. Miller et al 2004 doi:10.1029/2003JD004085 or Miller et al 2014 doi:10.1007/978-94-017-8978-3_13)

We have added the sentence(P29 L621):

"One mechanism for this is the perturbation of the hydrological cycle by the dust DRE (e.g. Miller et al., 2004; Wilcox et al., 2010; Woodage and Woodward, 2014; Miller et al., 2014)."

533 "(haboobs)" Also note that this fraction will vary seasonally and be most important in summer (Caton-Harrison et al. JGR 2019 and 2020).

We have added the clause (P28 L597):

"particularly in summer when this fraction is most important (Caton-Harrison et al., 2019 and 2020)"

536 "The omission of dust aging " what mechanisms of aging are referred to here? What effect does this have on particle lifetime?

This has been changed to (P28 L601):

"The omission of dust ageing through chemical processing has the effect of increasing lifetime and therefore increasing dust concentrations remotely from sources".

Response to comments by Anonymous Referee 2 #:

**Major comments**:

1. The results in the paper may to some extent depend on the tuning strategy, which is not described in the manuscript. Please provide some details on this aspect.

Section 3.3 "Dust Tuning" has been rewritten to give more details of the tuning strategy.  See text after Ron Miller's major comment (1) above.

2. the fact that the results for the changes dust radiative effects are small is largely due to the cancellation of positive and negative effects, so it would be useful to also give minimum/maximum values or ranges of changes that occur with the changed design.

This section has been revised to include maximum and minimum values (P15 L387):

"In UKESM1 dust is generally more reflective than the surface in the shortwave (SW), except over ice and the brightest deserts, with the result that the net downward shortwave at the top of the atmosphere (ToA) is reduced everywhere but over these very light surfaces.  The global mean ToA SW dust DRE is -0.280 W m$^{-2}$, and -0.410 W m$^{-2}$ in the clearsky (CS).  The ToA longwave (LW) DRE is positive everywhere, with higher values over areas of higher load, particularly the Sahara.  It has a global mean of 0.194 W m$^{-2}$ (0.237 W m$^{-2}$ CS).  SW and LW combine to give a positive net DRE over the Sahara with a maximum of 3.71 Wm$^{-2}$, but partially cancel in most other regions to produce modest positive net values over lighter surfaces and elsewhere negative net values down to -3.05 Wm$^{-2}$ in the Atlantic under the Saharan plume, giving a global mean of only -0.086 W m$^{-2}$.  At the surface the global mean net DRE is -0.168 W m$^{-2}$, being positive over the brightest surfaces and negative elsewhere, with a maximum of 4.45 Wm$^{-2}$ in the Sahara and a minimum of -3.19 Wm$^{-2}$ under the Saharan plume.  The surface SW DRE is negative everywhere and has a global mean of -0.556 W m$^{-2}$ (-0.679 W m$^{-2}$ CS); the LW is positive everywhere, with a global mean of 0.388 W m$^{-2}$ (0.455 W m$^{-2}$ CS).

In HaGEM3-GC3.1 the global mean net DRE at ToA is -0.269 W m$^{-2}$, varying between -7.18 W m$^{-2}$ and 1.90 W m$^{-2}$; and at the surface the mean is -0.350 W m$^{-2}$, with a range from -5.75 W m$^{-2}$ to 2.88 W m$^{-2}$.  The larger shortwave and smaller longwave effect are associated with the difference in size distribution between the UK_PD and H3_PD simulations, as will be explored in Section 5.6."

3. Please adapt the number significant digits given in the results to reflect their uncertainty range. E.g. three significant digits for the radiative effects clearly overstates its accuracy.  Overall, some discussion on model uncertainties and variabilities should be provided to put the results of this work in context.

We accept that the number of significant figures used does not reflect the uncertainty associated with the model estimates of real-world dust effects.  However, we are here reporting the results of particular model simulations - values which are known to a high degree of accuracy; and in our view the use of three significant figures provides useful information by allowing the reader to see how some of the small difference terms were obtained.  This approach is fairly common practice in dust modeling studies (eg Balkanski et al, 2007; Huneeus et al, 2011; Albani et al, 2014; Scanza et al 2016; etc).  The discussion section has been re-written and extended to include more information and comments on uncertainties.  The full text of this section is given in response to comment 15 below.

**Specific comments:**

1. Equation 1 (page 3) – please provide units for the variables -is the factor D dimensionless?

SI units are used throughout.  We have now added information on units., and state that D is dimensionless (P3 L85):

"…where $\rho$ is surface air density in kg m$^{-3}$, B is bare soil fraction, $U^*_{ti}$ is threshold friction velocity for the bin in m s$^{-1}$, $U^*$ is friction velocity excluding orographic effects in m s$^{-1}$, C is a constant of proportionality set to 2.61 from wind-tunnel experiments, D is a dimensionless tuneable parameter (see Section 3.3) and g is acceleration due to gravity in m s$^{-2}$.  $M_i$ is the mass fraction of particles in the bin, obtained from soil clay, silt and sand fractions from HWSD data (Nachtergaele et al., 2008) according to the method described in Woodward (2001)"

Page 4+5 – various dimensionless tunable constants are mentioned (D, k1, k2) – please provide the actual values, maybe in a table.

The values of all the tunable constants are given in section 3.3 "Dust tuning".  This was referenced after eqn (2) as "The value of $k_1$ was chosen empirically, as described in section 3.3", and we have now added references in the paragraphs after eqn (1)  (see response to previous comment)) " and before eqn (4) (P4 L120):

"a dimensionless tunable constant $k_2$, which was also set empirically (see Section 3.3)"

2.    Which datasets are used for the input data used to describe soil properties?

Information on soil parameters is given in Section 2: "The soil parameters also differ: those in HadGEM3-GC3.1 are based on Van Genuchten (Loveland et al., 2000), whilst those in UKESM1 are from Brooks and Corey (1964)."

3.    Line 118 - please provide the limits for the size bins

The limits for the vertical flux bins are given near the start of this section: " Dust emission in six size bins with boundaries at 0.06324, 0.2, 0.6324, 2.0, 6.324, 20.0 and 63.24 μm diameter is calculated at each atmospheric model timestep (20 mins)." We have reworded the sentence ( at former l 118) for clarity (P5 L133):

"The vertical dust flux is calculated for the six emission bins in the range 0.06324 to 63.24 μm"

4.    Line 136 – in which regions are seasonally varying dust sources occurring?

This can now be seen in the difference between the source areas and bare soil areas in H3_PD, shown a new Fig 10.  This is referenced at this point in the text (P5 L154) :

"Seasonal sources (see Fig 10.) accounted for less than 10% of the load in HadGEM3-GC3.1, so this was not expected to have a large impact".

[Figure]

**Figure 10: Grass fraction, tree and shrub fraction, bare soil fraction and dust source area fraction from UK_PD, UK_SSP5, H3_PD and H3_SSP5.  (The H3_PD and H3_SSP5 source areas include seasonal sources.)**

5.    Overall, Section 3.1 would benefit from a discussion of uncertainties

We address uncertainties in the emission scheme in a newly extended section in the Discussion (see response to major comment (3)), which is now referenced at the end of 3.1 as (P5 L155)

"The uncertainties associated with the emission scheme are considered in Section 7".

6. Line 195 – Dust load results are compared to AEROCOM means from Textor et al (2006) – more recent AEROCOM results that should be used as benchmark are found e.g. in: Gliss et al. 2021 (https://acp.copernicus.org/articles/21/87/2021/acp-21-87-2021.pdf)

This paragraph been amended to include data from the new reference (P8 L235):

"Figure 1 shows dust load from the UKESM1 and HadGEM3-GC3.1 present-day simulations (UK_PD and H3_PD), together with the differences and fractional differences. Both models capture the expected global dust distributions qualitatively well. The UK_PD global load of 19.5 Tg is 30% higher than the 15.0 Tg H3_PD value. These values span the AeroCom phase III mean of 16.6 Tg, and are within the range of 5.7 Tg to 22.3 Tg shown by participating models (Gliss et al., 2021). They are also consistent with the more observationally constrained estimate of Kok et al. (2017) for PM20 dust load of 23 Tg with a range from 14 Tg to 33 Tg. The greatest differences between the models are over the Sahel, India, the Middle East, Asian midlatitudes and Australia, and are predominantly due to differences in the bare soil and associated changes (see Section 5.6 and Fig. 8)."

7. Figure 1 compares results from the different model versions, they also differ in their treatment of seasonal sources. It would be interesting to see the regional effect of turning on/off seasonal sources, even if the overall impacts of seasonality are small.

As the results of experiments H3_TUK_INSS and H3_TUK_EXSS showed that the impact of excluding seasonal sources was relatively small, no action was taken to allow seasonal sources to be represented in UKESM1 (as this would not have been a trivial task). This work focusses on dust in UKESM1 and investigation of the effect of seasonal sources in HadGEM3-GC3.1 is outside its scope. However, the locations of the seasonal emissions can now be gauged from comparison of the seasonal source areas of H3_PD derived from the new Fig. 10 (above) with the emissions areas shown in the new Fig. 12 (below).

8. Section 5.3 – In addition to the comparison of AERONET AODs I suggest to also compare Angstrom parameters from the AERONET data as well – these should provide additional information for the particle sizes.

We have followed this suggestion, and a new paragraph has been added to section 5.3 (P11 L301)

"The observed multi-annual mean AOD at 440 nm meaned over these AERONET sites is 0.45, whilst UK_PD simulates 0.28 and H3_PD simulates 0.37. Comparison of the simulations shows that the lower AOD in UK_PD is due to a combination of lower dust optical depth (DOD), (0.13 in UK_PD and 0.19 in H3_PD) with lower optical depth due to other species (0.15 in UK_PD and 0.18 in H3_PD). The lower DOD in UK_PD is caused by the larger particle sizes in that model, as evidenced by the Angstrom exponent for the DOD (440-870) which is -0.06 in UK_PD and 0.23 in H3_PD, whilst the mean atmospheric load at these sites is higher in UK_PD: 540 mg m$^{-2}$, compared with 340 mg m$^{-2}$ in H3_PD. The Angstrom exponent for the total AOD (440-870) is 0.65 in UK_PD and 0.76 in H3_PD, compared with 0.33 in the AERONET observations. This indicates too low a coarse mode fraction in the simulations which suggests too little dust, as this is the dominant coarse mode species at these locations and the low simulated AODs show that there is no excess of fine mode aerosol. The slightly better agreement of UK_PD with the observed Angstrom exponent may be an indication that the particle size distribution is more realistic than in H3_PD, which would be consistent with the comparisons with FENNEC data, though with the caveat that the potential effects of different concentrations of other aerosol species at the AERONET sites cannot be ignored."

9. Table 2 on page 15 summarizes key results but is difficult to read - possibly separating into two tables for absolute values and percentages, respectively, would help? Also, for TOA the forcing values over land and ocean would differ and partly cancel each other, so it would be useful to list them separately.

Partial cancellation does indeed occur in the calculation of global mean ToA DREs. However, as can be seen in Fig. 7, the cancellation is not due to land/ocean differences in sign. Fig. 7e shows the UK_PD ToA LW component is positive everywhere, whilst Fig. 7a shows the ToA SW is negative everywhere except the brightest land surface: parts of the Sahara, the Arabian peninsula, Greenland and Antarctica. This results in a net ToA (Fig. 7i) which is negative over most of the land but positive over (larger) areas of the Sahara, Arabian Peninsula, Greenland and Antarctica, and also parts of Australia and southern Africa. Over the ocean the net ToA in UK_PD (Fig. 7i) is also mostly negative, though positive over the Southern Ocean, and much of the Arctic. Thus there are areas of both positive and negative net ToA forcing, which partially cancel over both land and ocean. We therefore think that separating the forcing into land and ocean components would not be helpful in this case. Table 2 has been reformatted in an attempt to make it more easily readable:

| Experiment | Load (Tg) | ToA SW (W m⁻²) | ToA LW (W m⁻²) | ToA Net (W m⁻²) | Surf.SW (W m⁻²) | Surf. LW (W m⁻²) | Surf. Net (W m⁻²) |
|---|---|---|---|---|---|---|---|
| H3_PD | 15.01 | -0.460 | +0.164 | -0.296 | -0.688 | +0.338 | -0.350 |
| (1) Difference due to tuning (size distribution) | - | +0.276 [ -60] | -0.021 [ -13] | +0.255 [ -86] | +0.288 [ -42] | -0.026 [ -8] | +0.262 [ -75] |
| H3_TUK_INSS_NL | 15.01 | -0.184 | +0.143 | -0.041 | -0.400 | +0.312 | -0.088 |
| (2) Difference due to tuning (load) | -4.63 [ -31] | +0.057 [ -31] | -0.044 [ -31] | +0.013 [ -31] | +0.123 [ -31] | -0.096 [-31] | +0.027 [ -31] |
| H3_TUK_INSS | 10.38 | -0.127 | +0.099 | -0.028 | -0.277 | +0.216 | -0.061 |
| (3) Difference due to seasonal sources | -0.98 [ -9] | +0.011 [ -9] | -0.009 [ -9] | -0.002 [ -9] | -0.027 [ -10] | -0.020 [ -9] | +0.007 [ -12] |
| H3_TUK_EXSS | 9.40 | -0.116 | +0.090 | -0.026 | -0.249 | +0.196 | -0.053 |
| (4) Difference due to driving model | +10.14 [+108] | -0.164 [+141] | +0.103 [+115] | -0.061 [+236] | -0.307 [+123] | +0.192 [+98] | -0.115 [+215] |
| UK_PD | 19.54 | -0.280 | +0.194 | -0.086 | -0.556 | +0.388 | -0.168 |
| Difference from H3_PD to UK_PD | +4.52 [ +30] | -0.180 [ -39] | -0.030 [ +18] | -0.210 [ -71] | -0.132 [ -19] | -0.049 [+15] | -0.182 [ -52] |

Table 2: Global mean load and all-sky direct radiative effects due to dust in a present-day climate, from various experiments. Alternate rows show experimental results and the absolute differences, followed by percentage differences in brackets, between the experiment below and the experiment above.

10. Line 363 – please provide information on the quantitative change of the bare-soil source area.

We have added information on bare soil area (P18 L434):

"The change in driving model (Factor 4) more than doubles the load and produces much the largest individual impact on the atmospheric burden. The most important element in this change is the vegetation: the disparity between the bare soil simulated by TRIFFID in UKESM1 and the IGBP climatology used in HadGEM3-GC3.1 accounts for 70% of the extra emissions in UK_PD compared with H3_TUK_EXSS. The global average bare soil fraction is 0.26 in UK_PD and 0.24 in H3_UK_EXSS, but regional variations are much larger and their geographic distribution promotes extra dust production in UK_PD. The 29% of the UK_PD bare soil area that is vegetated in H3_TUK_EXSS is mostly in arid areas at the margins of existing deserts where conditions favour dust emission, whilst the 25% of the H3_TUK_EXSS bare soil area that is vegetated in UK_PD is mainly in regions where moisture limits dust emission (Fig. 8g)"

11. Line 363-365 – the effects from vegetation changes on meteorology and soil moisture are a central new result of this work, the description of these effects should be more detailed.

We see this effect in the difference between the UK_PD and H3_UK_EXSS simulations – i.e. as a consequence of different driving models. It would be most interesting to investigate this further if it appeared as a response to changing climate, however that does not occur to a significant extent in our simulations because there is so little vegetation loss under a warming climate and increasing $CO_2$. We have added some more detail to the description here (P18 L442), and also mentioned it in the Discussion (P29 L633):

"Areas where there is less vegetation and thus reduced roughness might be expected to be associated with higher near-surface wind speed and hence increased evaporation and reduced soil moisture, providing conditions particularly favourable to dust production. This can be seen in the regions of greatest difference between UK_PD and H3_TUK_EXSS such as the Sahel, India and the Kazakh Steppe (Fig. 8g,d,k,o)"

"Also, where vegetation is lost not only do source areas increase, but the consequent reduced roughness may lead to increased windspeed, and hence via increased evaporation to reduced soil moisture - changes which tend to enhance dust emissions.

However, this effect is small in our climate change studies as vegetation tends to grow rather than die back in response to increased $CO_2$ and a warmer climate."

12. Line 372 – is the change in modelled size distribution due to different emission schemes or datasets or due to a different meteorology?

We have reworded this sentence to clarify (P19 L450):

"Change in size distribution consequent on re-tuning (Factor 1) is the factor which produces the largest absolute change in the net DREs…."

13. Line 401 "all the processes in UKESM1" – please elaborate

We have reworded this as (P21 L482)

"due to other model differences, including the impact of the extra ESM processes and feedbacks included in UKESM.1"

14. Figure 10+11 can hardly be read due to the small panels – it could be split up in several parts. Also consider to move them to an Appendix.

We have re-arranged these two figures into three, to increase the size of the panels in the printed version of the paper. We now show the vegetation changes from both models in a new Fig 10 (see above), and the dust and driving field changes from UKESM1 and from HadGEM3-GC3.1 in new Figs 11 and 12 respectively. We think these results are too central to the paper to be moved to an appendix.

[Figure]

**Figure 11: Soil moisture in top layer, dust source area fraction, 10m windspeed, dust emission diagnostic and dust load from UK_PD and UK_SSP5, together with differences and fractional differences between experiments.**

[Figure]

**Figure 12: as Fig. 11 but for H3_PD and H3_SSP5.**

15. Discussion: Page 25 – here issues are mixed. The discussion on differences in the dust size distribution should be clearly separate from discussion of the effects of computing dust in an Earth system model with interactive vegetation.

The discussion has been extensively re-written and expanded, and these two factors are now treated more independently. However, the discussion is structured to broadly follow the layout of the paper with a section on the present-day simulation followed by a section on the response to climate change, and this is followed by comments on model uncertainties. Within this framework it is impossible to separate out discussion of size and vegetation completely. The section now read as follows:

[revised manuscript text omitted]

**Typos**

1. Line 231 Remove double period at end of sentence
2. Line 233 dimeter -> diameter

These have been corrected.